# Downstream Task Guided Masking Learning in Masked Autoencoders Using Multi-Level Optimization

**Han Guo**                                                    *h5guo@ucsd.edu*
*UC San Diego*

**Ramtin Hosseini**                                           *rhossein@ucsd.edu*
*UC San Diego*

**Ruiyi Zhang**                                               *ruz048@ucsd.edu*
*UC San Diego*

**Sai Ashish Somayajula**                                     *ssomayaj@ucsd.edu*
*UC San Diego*

**Ranak Roy Chowdhury**                                       *rrchowdh@ucsd.edu*
*UC San Diego*

**Rajesh K. Gupta**                                           *rgupta@ucsd.edu*
*UC San Diego*

**Pengtao Xie**[†]                                            *p1xie@ucsd.edu*
*UC San Diego*

Reviewed on OpenReview: *https://openreview.net/forum?id=cFmmaxkD5A*

## Abstract

Masked Autoencoder (MAE) is a notable method for self-supervised pretraining in visual representation learning. It operates by randomly masking image patches and reconstructing these masked patches using the unmasked ones. A key limitation of MAE lies in its disregard for the varying informativeness of different patches, as it uniformly selects patches to mask. To overcome this, some approaches propose masking based on patch informativeness. However, these methods often do not consider the specific requirements of downstream tasks, potentially leading to suboptimal representations for these tasks. In response, we introduce the Multi-level Optimized Mask Autoencoder (MLO-MAE), a novel framework that leverages end-to-end feedback from downstream tasks to learn an optimal masking strategy during pretraining. Our experimental findings highlight MLO-MAE's significant advancements in visual representation learning. Compared to existing methods, it demonstrates remarkable improvements across diverse datasets and tasks, showcasing its adaptability and efficiency. Our code is available at `https://github.com/Alexiland/MLO-MAE`

## 1 Introduction

In the rapidly evolving field of self-supervised learning (Balestriero et al., 2023; Gui et al., 2023), particularly in visual representation learning, Masked Autoencoder (MAE) (He et al., 2022) has emerged as a prominent approach, which draws inspiration from the successful masked language models like BERT (Devlin et al.,

---

[†]Corresponding author

2018) and RoBERTa (Liu et al., 2019). Similar to how BERT learns textual representations by predicting randomly masked tokens, MAE is designed to learn visual representations by masking random patches of an image and then reconstructing them using the remaining unmasked ones.

Although MAE has shown empirical success, it applies a uniform random approach to mask patches, overlooking the varying distribution of information across different image regions (Chen et al., 2023; Kong & Zhang, 2023; Wang et al., 2023; Liu et al., 2023). It assumes equal informativeness across all parts of an image, an assumption that does not always hold true. Some image areas may hold more critical information than others, a factor not considered in MAE's current design. Such oversight might hinder the model's capability and efficiency in learning representations. This lack of distinction between more and less informative regions in the MAE could lead to disproportionate allocations of computational resources. Consequently, the model might spend excessive effort on less significant areas while inadequately processing and capturing the nuances in regions that contain more valuable information.

To mitigate this limitation, various strategies have been suggested for masking patches contingent on their informativeness. Key approaches include masking regions with high attention scores to prioritize areas of interest (Li et al., 2021; Kakogeorgiou et al., 2022); employing semantic segmentation to identify and mask regions rich in information (Li et al., 2022); automatically learning a masking module (Madan et al., 2024); and learning a differentiable mask generator via adversarial training (Chen et al., 2023). These approaches aim to refine the masking process by prioritizing patches based on the level of information they contain, rather than treating all patches uniformly.

Although these methods are promising, they mask patches without incorporating feedback from downstream tasks. Their process involves two separate stages: initially employing a specific masking strategy to pretrain an image encoder, then using this encoder to perform downstream tasks (via finetuning (He et al., 2022) or linear probing (He et al., 2022)), with the hope that the encoder pretrained using this strategy will be effective for these tasks. During this process, the design of the masking strategy is not influenced by the requirements of the downstream tasks. As a result, the representations developed through this strategy may not be well-aligned with the needs of these tasks, which could limit their effectiveness.

To bridge this gap, we propose a *downstream task guided* masking strategy learning framework based on multi-level optimization (MLO) (Vicente & Calamai, 1994). Our approach utilizes feedback from downstream tasks to autonomously learn the optimal masking strategy. It pretrains an image encoder and applies the pretrained encoder to perform a downstream task in an end-to-end manner, allowing the downstream task's performance to directly influence the masking process during pretraining. Our method learns a masking network to mask patches. It processes an input image to identify specific patches for masking. Our method consists of three interconnected stages. In the first stage, a preliminary version of the masking network masks certain patches, followed by the pretraining of an image encoder tasked with reconstructing these masked patches. In the second stage, we utilize this encoder to construct a downstream model, which is subsequently trained using the training dataset specific to a downstream task. The final stage involves evaluating the downstream model using a held-out validation dataset. The effectiveness of the masking network is indirectly measured by the downstream model's validation performance. An inferior masking network might fail to correctly identify the optimal patches for masking, leading to ineffective pretraining of the image encoder. When applied to the downstream task, the encoder's inadequate representation learning capabilities will lead to suboptimal validation performance of the downstream model. To prevent this, we continuously refine the masking network, ensuring it maximizes downstream validation performance. Each of these stages is formulated as one level of optimization problem in our MLO framework. The three levels of optimization problems are mutually dependent on each other and solved jointly. This enables the three stages to be conducted end-to-end, where the downstream validation performance closely guides the learning of the masking network.

The major contributions of this work include:

- We propose a multi-level optimization based end-to-end framework to learn an optimal masking strategy in Masked Autoencoder by leveraging feedback from downstream tasks.

- Our approach outperforms a range of leading-edge methods in learning representations, as evidenced across various datasets such as CIFAR-10, CIFAR-100, and ImageNet-1K.

- Our method showcases remarkable transfer learning abilities, in fine-grained classification, semantic segmentation, and object detection tasks, demonstrated on datasets including CUB-200-2011, Stanford Cars, iNaturalist 2019, ADE20K, and MS-COCO.

## 2 Related works

### 2.1 Masked autoencoders

Following the success of masked language models in the field of natural language processing (Devlin et al., 2018), various masked image models have been proposed (Chen et al., 2020; Bao et al., 2022). Among them, Masked Autoencoder (MAE) has become a promising methodology for generic visual pretraining (He et al., 2022). MAE is a denoising autoencoder that randomly masks the input image and tries to reconstruct the missing pixels. It uses a high masking ratio (75% in MAE compared to 15% in BERT) and a lightweight decoder architecture that forces the encoder to learn meaningful visual representations. Zhang et al. (2022) propose a theoretical framework to understand the role of masking in MAE, and introduce a Uniformity-enhanced MAE (U-MAE) to address the dimensional collapse issue. Despite MAE's effectiveness, recent works underscore the importance of replacing the random patch masking method in MAE with more sophisticated masking strategies (Kakogeorgiou et al., 2022; Shi et al., 2022). For instance, MST (Li et al., 2021) utilizes attention maps to guide the masking process, selectively obscuring less attended regions to maintain important information. SemMAE (Li et al., 2022) combines a StyleGAN-based decoder with the MAE decoder and leverages attention maps from the StyleGAN decoder to provide semantic cues for patch masking. Furthermore, some recent methods propose to use a learnable masking module to generate masking strategies and optimize the masking module in the pretraining process. For example, AutoMAE (Chen et al., 2023) links a differentiable mask generator with MAE using Gumbel-Softmax (Jang et al., 2016), following a similar two-stage setup as in SemMAE. CL-MAE (Madan et al., 2024) leverages curriculum learning to enhance MAE by progressively increasing the complexity of the masks generated from a learnable masking module. Compared to these existing methods, the key distinction of our approach lies in the utilization of feedback from downstream tasks to inform the development of masking strategies, a mechanism absent in the current methodologies.

### 2.2 Bi-level and multi-level optimization

Recently, Bi-level Optimization (BLO) and Multi-level Optimization (MLO) techniques have been widely applied for meta-learning (Feurer et al., 2015; Finn et al., 2017), neural architecture search (Cai et al., 2019; Xie et al., 2019; Xu et al., 2020; Hosseini et al., 2021) and hyperparameter tuning (Feurer et al., 2015; Baydin et al., 2017). BLO, a formulation that consists of two levels of nested optimization problems, has been broadly applied in numerous machine learning applications (Liu et al., 2018; Liang et al., 2019). BLO based methods have enabled automatic and efficient learning of upper-level parameters, such as meta parameters and neural architectures, thereby reducing the need for extensive hyperparameter tuning through manual efforts. Following the success of BLO, MLO - which has more than two levels of nested optimization problems (Hosseini & Xie, 2022; Hosseini et al., 2023; Sheth et al., 2021; Garg et al., 2021) - has been used to solve machine learning tasks with more complicated dependencies. These works develop multi-stage pipelines, with each stage corresponding to one level of optimization problem (OP). Different stages are executed end-to-end by solving all levels of interdependent OPs jointly. Despite its effectiveness, MLO based methods increase memory and computation costs due to their growing number of optimization levels. To tackle this challenge, Choe et al. (2022) develop software that integrates multiple approximation algorithms to efficiently compute the hypergradients within BLO and MLO problems.

## 3 Methods

### 3.1 Overview

We introduce the Multi-level Optimized MAE (MLO-MAE), an end-to-end visual representation learning method which leverages the guidance from a downstream task to learn an optimal masking strategy automatically. Conceptually, MLO-MAE reimagines the visual representation learning by coupling pretraining

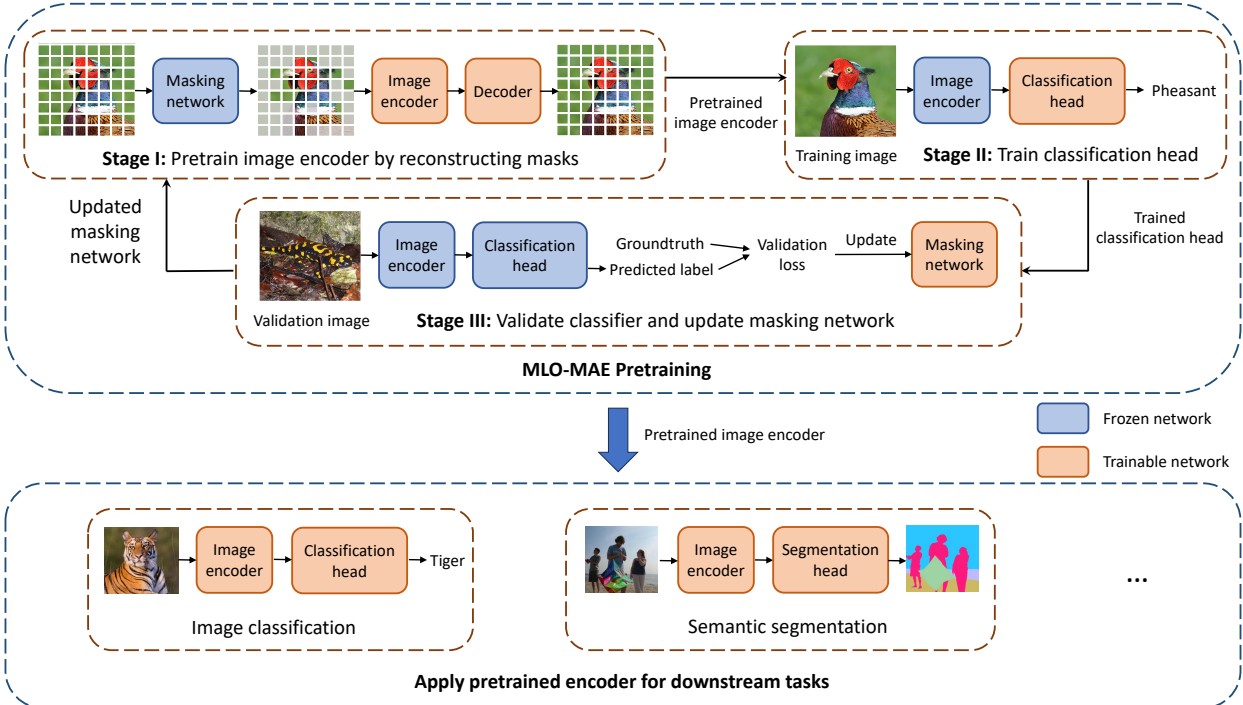

Figure 1: An overview of MLO-MAE, which consists of three stages performed end-to-end. Modules with learnable parameters are indicated in orange, and those with frozen parameters are in blue.

with downstream task performance. MLO-MAE introduces a learnable masking strategy that prioritizes patches based on their relevance to a downstream task. This enables the model to focus its representation learning on features that are most critical for achieving better downstream performance.

As illustrated in Figure 1, the architecture of MLO-MAE consists of three key components: a Vision Transformer (ViT) (Dosovitskiy et al., 2020)-based image encoder $E$, a classification head $C$, and a masking network $T$. The masking network processes input images to identify patches for masking. The image encoder then extracts a representation by reconstructing the masked patches using information from the unmasked ones, serving as the backbone for visual representation learning. The classification head $C$ predicts a class label based on the image representation extracted by the encoder. The pretraining process is carried out on an unlabeled dataset $\mathcal{D}_u$, similar to other self-supervised methods. However, the learning of the masking network is guided by a downstream image classification task using a labeled dataset $\mathcal{D}$, which is further divided into a training subset $\mathcal{D}^{tr}$ and a validation subset $\mathcal{D}^{val}$.

MLO-MAE operates in three interconnected stages. In the Stage I, the masking network $T$ generates a preliminary mask for input images from $\mathcal{D}_u$, and the image encoder $E$ is pretrained on these masked images by minimizing reconstruction loss. In the Stage II, the pretrained encoder extracts representations for images in the training subset $\mathcal{D}^{tr}$. These representations, along with their associated labels, are used to train the classification head $C$ by minimizing classification loss. Finally, in Stage III, the pretrained encoder extracts representations for the validation subset $\mathcal{D}^{val}$, which are passed to the classification head for label predictions. Validation loss, computed by comparing the predicted labels with the actual labels, serves as a feedback mechanism for evaluating the effectiveness of the masking network $T$. By focusing on minimizing this validation loss, the masking network $T$ is iteratively refined.

To seamlessly integrate these stages, MLO-MAE employs a multi-level optimization (MLO) framework, where each stage corresponds to one level of optimization. Optimal parameters obtained at each lower level serve as inputs for the loss functions at the subsequent upper levels. Conversely, non-optimal parameters from the upper levels are utilized to define the loss functions at lower levels. These nested optimization problems are solved jointly, enabling the three stages to operate in an end-to-end manner. This multi-level

approach ensures that the pretraining process dynamically aligns with the downstream task requirements, allowing MLO-MAE to learn a masking strategy that optimally enhances visual representation learning.

### 3.2 Multi-level optimization framework

The framework of our proposed MLO-MAE is structured into three interconnected stages. These three stages are integrated within a multi-level optimization framework.

**Stage I: pretrain image encoder.** Given an input image $X \in D_u$ divided into $N$ non-overlapping patches of equal size, denoted as $\{P_i\}_{i=1}^N$, the masking network $T$ takes $X$ as input and generates a probability $\sigma(P_i, X; T)$ for each patch $P_i$, which represents the likelihood that $P_i$ should be masked. Given a masking ratio $r$, a hyperparameter dictating the proportion of patches to be masked, we first rank all patches in descending order based on their masking probabilities. We then select the top $N \times r$ patches denoted as $\mathcal{M}(X; T, r)$, those with the highest probabilities, and mask them. The remaining patches, denoted as $X - \mathcal{M}(X; T, r)$, are unmasked. Then we feed the unmasked patches into an autoencoder (He et al., 2022), which consists of the image encoder $E$ and a decoder $D$, to reconstruct the masked patches $\mathcal{M}(X; T, r)$. In detail, the unmasked patches are first processed by the image encoder $E$, which is responsible for extracting their representations. Then, these representations are input into the decoder $D$. The decoder's role is to accurately predict the pixel values of the masked patches. To evaluate the performance of this reconstruction, we employ a reconstruction loss, $\mathcal{L}_{rec}$, defined as the squared differences between the predicted and ground truth pixel values of the masked patches. Importantly, the reconstruction loss for the $j$-th masked patch $\mathcal{M}_j(X; T, r)$ is weighted according to its masking probability $\sigma(\mathcal{M}_j(X; T, r), X; T)$. This probability reflects the likelihood of a patch being masked and thus, guides the autoencoder to prioritize the reconstruction of patches deemed more likely to be masked.

In this stage, we provisionally hold the masking network $T$ constant, and focus on training the image encoder $E$ and decoder $D$, by solving the following optimization problem:

$$E^*(T), D^* = \operatorname*{argmin}_{E,D} \sum_{X \in \mathcal{D}_u} \sum_{j=1}^{N \times r} \sigma(\mathcal{M}_j(X; T, r), X; T) \mathcal{L}_{rec}(X - \mathcal{M}(X; T, r), \mathcal{M}_j(X; T, r); E, D). \tag{1}$$

The notation $E^*(T)$ indicates that the optimal solution $E^*$ is a function of $T$, as $E^*$ is determined by the loss function which in turn depends on $T$.

**Stage II: train classification head.** Utilizing the pretrained image encoder $E^*(T)$ from Stage I, we develop an image classification model for a downstream task. This model comprises the encoder $E^*(T)$ and the classification head $C$. For any given input image, it is first processed by the encoder to generate a representation. This representation is then input into the classification head to determine the class label. In this stage, we keep the encoder parameters fixed and focus on training the classification head. This is achieved by minimizing a cross-entropy classification loss $\mathcal{L}_{cls}$ on the training dataset $\mathcal{D}^{tr}$:

$$C^*(E^*(T)) = \operatorname*{argmin}_{C} \mathcal{L}_{cls}(\mathcal{D}^{tr}; E^*(T), C). \tag{2}$$

**Stage III: update masking network.** In Stage III, we assess the classification model developed in Stage II on the validation set $\mathcal{D}^{val}$. This model integrates the image encoder, $E^*(T)$, which was pretrained in Stage I, and the classification head, $C^*(E^*(T))$, trained in Stage II. The validation loss serves as an indirect measure of the efficacy of the masking network $T$. Our objective is to enhance the performance of $T$ by minimizing this validation loss:

$$\min_T \mathcal{L}_{cls}(\mathcal{D}^{val}; E^*(T), C^*(E^*(T))). \tag{3}$$

**Multi-level optimization.** Integrating the three optimization problems together, we have the following multi-level optimization problem:

$$\min_T \mathcal{L}_{cls}(\mathcal{D}^{val}; E^*(T), C^*(E^*(T)))$$

$$s.t.\ C^*(E^*(T)) = \operatorname*{argmin}_{C} \mathcal{L}_{cls}(\mathcal{D}^{tr}; E^*(T), C)$$

$$E^*(T), D^* = \operatorname*{argmin}_{E,D} \sum_{X \in \mathcal{D}_u} \sum_{j=1}^{N \times r} \sigma(\mathcal{M}_j(X; T, r), X; T) \mathcal{L}_{rec}(X - \mathcal{M}(X; T, r), \mathcal{M}_j(X; T, r); E, D) \tag{4}$$

Table 1: Top-1 accuracy (%) on the test sets of CIFAR-10, CIFAR-100, and ImageNet, in fine-tuning experiments. The baseline methods SemMAE and AutoMAE are not included in the comparison on CIFAR-10 and CIFAR-100, due to the absence of reported results for these datasets in their original publications and the unavailability of their implementation code for conducting evaluations on these datasets.

| | (No Pretraining) | (Random Masking) | | (Learnable Masking) | | |
| | ViT | MAE | U-MAE | SemMAE | AutoMAE | MLO-MAE (Ours) |
|---|---|---|---|---|---|---|
| CIFAR-100 | 56.4 | 64.0 | 64.6 | – | – | **79.4** |
| CIFAR-10 | 82.3 | 93.7 | 94.3 | – | – | **96.2** |
| ImageNet-1K | 77.9 | 83.6 | 83.0 | 83.3 | 83.3 | **84.8** |

In this formulation, the three levels of optimization problems are mutually dependent. The first level's output, $E^*(T)$, defines the loss function in the second level. Both the outputs of the first and second levels are fed into the loss function of the third level. Simultaneously, the third level's optimization variable, $T$, influences the loss functions in the first two levels. By concurrently solving these optimization problems across all three levels, we enable an integrated, end-to-end execution of the three stages.

**Optimization algorithm.** Inspired by Liu et al. (2018), we develop an efficient hypergradient-based method to solve the problem in Eq.(4). First, we approximate the optimal solutions $E^*(T)$ and $D^*$ by executing several iterations (termed as unrolling steps) of gradient descent updates of $E$ and $D$ against the loss function at the first level. The approximation of $E^*(T)$ is then plugged into the second-level loss, and $C^*(E^*(T))$ is similarly approximated using multiple steps of gradient descent updates of $C$ against this approximate loss. The approximations of $E^*(T)$ and $C^*(E^*(T))$ are then applied to the third-level loss, enabling the gradient descent update of $T$. This iterative process of updating continues until convergence is achieved. Details of this optimization algorithm are deferred to Appendix A and Algorithm 1.

## 4 Experiments

### 4.1 Experimental settings

**Model setup and hyperparameters.** In our method, the masking network is structured with multiple layers. Initially, there is a linear layer, where the input size is determined by the product of the number of patches (196 for ImageNet and 256 for CIFAR) and the embedding dimension (we used the patch embedding method in ViT, with a dimension of 768), and it has a hidden size of 512. This is followed by a ReLU layer. Next, there is another linear layer, which takes an input size of 512 and produces an output size equivalent to the number of patches. Finally, a sigmoid activation function is applied to the output to generate probabilities in the range of 0 to 1. Implementation details are described in Appendix B.2. Following MAE (He et al., 2022), an asymmetric ViT (Dosovitskiy et al., 2020) encoder-decoder architecture was used for mask reconstruction. Recognizing the constraints of computational resources, we primarily employed the ViT-B (Dosovitskiy et al., 2020) as the image encoder, ensuring a balance between efficiency and performance. The classification head consists of a single linear layer. It is intentionally made simple to focus on evaluating the effectiveness of the learned representations. The patch size was set to 2 for CIFAR-10 and CIFAR-100, and 16 for ImageNet. For all experiments, unless otherwise specified, we used the default mask ratio of 75% as suggested in MAE (He et al., 2022).

The number of unrolling steps in the algorithm for solving the MLO problem was set to 2. We employed the AdamW optimizer (Loshchilov & Hutter, 2017) with $\beta$ values of 0.9 and 0.95 for optimizing all parameters. The learning rates were set specifically for different components: $1e-4$ for the image encoder, and $4e-5$ for both the classification head and the masking network. We used a batch size of 256. For training, we set the epoch number to 50 for the ImageNet dataset and to 200 for the CIFAR datasets. All experiments were conducted on Nvidia A100 GPUs. Further information on our experimental settings can be found in Appendix B.

**Baselines.** We conducted comparisons with several baselines, including: 1) vanilla MAE (He et al., 2022) and U-MAE (Zhang et al., 2022), which employ uniform random masking of images; 2) SemMAE (Li et al., 2022) and AutoMAE (Chen et al., 2023), which mask patches according to their informativeness; and 3)

Table 2: Test accuracy (%) in linear probing experiments.

|  | MAE | U-MAE | SemMAE | AutoMAE | MLO-MAE (Ours) |
|---|---|---|---|---|---|
| CIFAR-100 | 46.6 | 50.4 | – | – | **63.8** |
| CIFAR-10 | 73.5 | 77.1 | – | – | **84.3** |
| ImageNet-1K | 68.0 | 58.5 | 65.0 | 68.8 | **70.2** |

Table 3: Accuracy (%) on fine-grained image classification datasets. All methods use ViT-B as the backbone with a patch size of 16.

| Method | iNaturalist | CUB | Cars |
|---|---|---|---|
| MAE | 79.5 | 83.3 | 92.7 |
| SemMAE | 79.6 | 82.1 | 92.4 |
| AutoMAE | 79.9 | 83.7 | 93.1 |
| MLO-MAE (Ours) | **80.1** | **84.0** | **93.4** |

Table 4: Semantic segmentation results on ADE20K.

| Method | mIoU |
|---|---|
| Supervised Pretraining | 45.3 |
| MAE | 48.1 |
| SemMAE | 46.3 |
| AutoMAE | 46.4 |
| MLO-MAE (Ours) | **49.8** |

the Vision Transformer (ViT) (Dosovitskiy et al., 2020) without pretraining by MAE methods (i.e., directly trained for classification from scratch). ViT-B was used as the image encoder in these methods. Following their original papers, the number of pre-training epochs for MAE, U-MAE, SemMAE, and AutoMAE on ImageNet are 1600, 200, 800, and 800 respectively. The patch size in all methods is 16 for ImageNet.

**Evaluation protocols.** In the literature on self-supervised learning, including MAE methods, there are two standard approaches for evaluating pretrained image encoders in downstream classification tasks (He et al., 2022). The first one is fine-tuning, which fine-tunes the pretrained encoder (together with training a randomly initialized classification head) by minimizing a classification loss on the downstream training data. The second approach is linear probing, which keeps the pretrained encoder fixed and only trains the classification head. Our experiments used both protocols. For each dataset $D$, pretraining was conducted on unlabeled images in $D$; fine-tuning and linear probing were conducted on $D$ as well, utilizing both images and their associated labels.

It is important to note that our method does not unfairly utilize more labeled data than the baselines. The labeled data used in Stage II and III of our framework is identical to that used in the fine-tuning phrase of the baselines.

### 4.2 Main results

**Fine-tuning results.** Table 1 shows the results. On the CIFAR-100 dataset, MLO-MAE demonstrates superior performance, achieving a test accuracy of 79.4%, substantially outperforming MAE's 64% and U-MAE's 64.6%. This trend of outperformance is also evident on the CIFAR-10 dataset, where MLO-MAE surpasses both MAE and U-MAE by 2.5% and 1.9% (absolute percentage) respectively. Moreover, on the ImageNet dataset, MLO-MAE performs better than all baseline methods. Specifically, MLO-MAE outperforms AutoMAE and SemMAE by 1.5% (absolute) improvements in top-1 accuracy.

These outcomes underscore MLO-MAE's strong capability in learning effective visual representations across datasets of varying scales, from the large-scale ImageNet to the smaller-sized CIFAR datasets. The superiority of MLO-MAE over MAE and U-MAE stems from its advanced masking strategy that selectively targets informative patches, a significant enhancement over the indiscriminate, random masking approach of the two baselines. Furthermore, MLO-MAE surpasses AutoMAE and SemMAE by integrating feedback from downstream classification tasks into its masking process. This dynamic adaptation contrasts with the static masking strategies of AutoMAE and SemMAE, which do not account for the specific requirements of downstream tasks, limiting their effectiveness.

**Linear probing results.** Table 2 shows the linear probing results. Our method MLO-MAE demonstrates superior performance compared to baselines across various datasets. Specifically, on ImageNet, MLO-MAE achieves an accuracy of 70.2%, substantially surpassing MAE's accuracy of 55.4% and U-MAE's 58.5%.

Table 5: Object detection result of MLO-MAE and baselines on MS-COCO detection task.

| Method | AutoMAE | MAE | MLO-MAE |
|---|---|---|---|
| $AP^{box}$ (%) | 50.5 | 50.3 | **51.1** |

Similarly, on the CIFAR-100 dataset, MLO-MAE continues to outperform, attaining an accuracy of 63.8%, significantly higher than the 46.6% accuracy of MAE and 50.4% accuracy of U-MAE.

The superiority of MLO-MAE compared to baseline methods stems from its unique approach of integrating the pretraining of the image encoder and linear probing in a seamless, end-to-end workflow. Specifically, MLO-MAE conducts pretraining at the first level and linear probing at the second level within a unified framework. This integration allows the linear probing performance, evaluated at the third level, to directly inform and enhance the pretraining process. Consequently, this leads to a pretrained encoder that is more effectively tailored for the downstream linear probing task. In contrast, baseline methods handle pretraining and linear probing as distinct, separate stages, where the performance of linear probing does not impact or contribute to the pretraining phase.

### 4.3 Transfer learning

### 4.3.1 Fine-grained image classification

In MLO-MAE, the masking network is trained using a specific downstream classification dataset, raising concerns about potential overfitting and limited generalizability to other datasets. To address this, we performed transfer learning experiments on labeled fine-grained classification datasets including iNaturalist 2019, CUB-200-2011, and Stanford Cars. Classification accuracy results, as presented in Table 3, show that MLO-MAE surpasses all baselines across these datasets. This indicates that the masking network, as learned by MLO-MAE for a particular downstream classification dataset, is capable of generalizing its effectiveness to additional datasets, rather than being overly tailored to that specific downstream classification dataset.

### 4.3.2 Semantic segmentation and object detection

We also explored the transferability of the masking network, initially learned through a downstream classification task, to other tasks including semantic segmentation and object detection. Given a ViT-B model pretrained on ImageNet using MLO-MAE or a baseline and subsequently fine-tuned on ImageNet, to transfer it for semantic segmentation, we integrated it as a backbone model into the UPerNet (Xiao et al., 2018) semantic segmentation framework. It was then further fine-tuned on the challenging ADE20K dataset (Zhou et al., 2017) containing 25K images spanning 150 semantic categories. The fine-tuning was conducted by the AdamW optimizer for over 160,000 iterations, with a batch size of 8 and a learning rate of 0.0001. In Table 4, MLO-MAE showcases a significant improvement over the baselines. Specifically, MLO-MAE attains enhancements in mean Intersection over Union (mIoU) by margins of 3.7% and 3.4% when compared to these baselines. This performance highlights the capability of MLO-MAE in executing dense prediction tasks.

Following MAE, we also adapt MLO-MAE pretrained ViT-B model for the use of an FPN backbone in Mask R-CNN. As shown in Table 5, MLO-MAE pretrained backbone model performs better than all baselines (51.1 comparing to 50.5 and 50.3, $AP^{box}$). We did not include SemMAE and U-MAE as they did not report on MS-COCO detection. Due to space limits, we defer the results of object detection on PASCAL VOC 2007 to Appendix D.1.

### 4.4 Continued pretraining

We further investigated the effectiveness of MLO-MAE in a continued pretraining setting. Starting with a ViT-B model pretrained by MAE on ImageNet, we applied MLO-MAE pretraining to 2 datasets, PDDB (Barbedo et al., 2018) and PAD-UFES (Pacheco et al., 2020). Table 6 compares the test accuracy (%) across three settings: (1) no continued pretraining, where the model is directly fine-tuned using labels; (2) continued pretraining on target dataset using MAE, followed by fine-tuning; (3) MLO-MAE pretraining solely on target dataset, followed by fine-tuning; and (4) continued pretraining on target dataset using MLO-MAE, followed by fine-tuning. Our results show that continued pretraining with MLO-MAE

Table 6: Continued pretraining on PDDB and PAD-UFES. All settings are initialized with weights of ViT-B pretrtained by MAE on ImageNet. Runtime measured in GPU hours on A100.

| Method | PDDB | | PAD-UFES | |
|---|---|---|---|---|
| | Acc(%) | Runtime | Acc(%) | Runtime |
| No continued pretraining | 88.6 | 2132 | 75.0 | 2132 |
| MAE continued pretraining | 89.3 | 2132 + 39 | 75.4 | 2132 + 3 |
| MLO-MAE pretraining (no IN pretrain) | 90.2 | 2132 + 37 | 75.9 | 2132 + 3 |
| MLO-MAE continued pretraining | **92.7** | 2132 + 37 | **77.6** | 2132 + 3 |

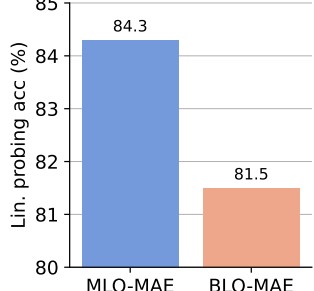

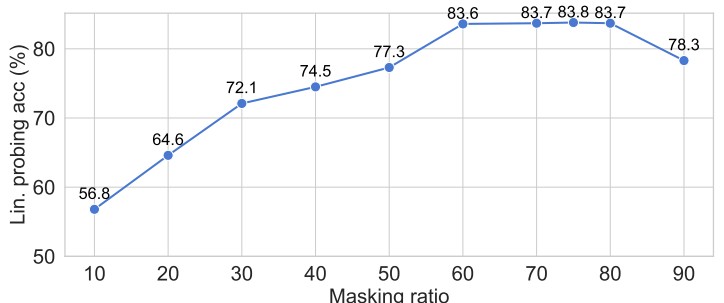

Figure 2: Comparison of BLO-MAE and MLO-MAE on CIFAR-10.

Figure 3: Impact of different masking ratios on linear probing performance on CIFAR-10.

significantly outperforms both MAE-based pretraining and no pretraining in both datasets. This highlights the practical advantage of MLO-MAE: by leveraging MAE pretraining once, subsequent tasks can benefit from fast, efficient continued pretraining with MLO-MAE, delivering substantial performance gains without requiring the computationally expensive process of large-scale pretraining from scratch for each downstream task.

### 4.5 Ablation studies

**Reduction to two levels.** To investigate the importance of maintaining three levels in the MLO-MAE framework, we simplified it to two levels, by combining the first and second levels, leading to the following bi-level optimization (BLO) problem (referred to as BLO-MAE):

$$\min_{T} \mathcal{L}_{cls}(\mathcal{D}^{val}; E^*(T), C^*)$$
$$s.t.\ E^*(T), D^*, C^* = \underset{E,D,C}{\operatorname{argmin}} \mathcal{L}_{cls}(\mathcal{D}^{tr}; E, C) + \gamma \sum_{X \in \mathcal{D}_u} \sum_{j=1}^{N \times r} \qquad (5)$$
$$\sigma(\mathcal{M}_j(X; T, r), X; T) \mathcal{L}_{rec}(X - \mathcal{M}(X; T, r), \mathcal{M}_j(X; T, r); E, D),$$

where $\gamma$ is a tradeoff parameter. Here, the image encoder $E$ is trained using a multi-task learning strategy, which involves minimizing the weighted sum of the pretraining loss and the downstream classification loss. Figure 2 presents the results. The BLO-MAE method leads to a notable decrease in accuracy by 2.8% compared to MLO-MAE. This highlights the significance of employing a three-stage process over a two-stage one. In the BLO-MAE approach, the lower level addresses a multi-task learning challenge by optimizing a weighted sum of losses from two distinct tasks. This scenario often leads to task competition, where minimizing the loss for one task inadvertently increases the loss for the other. Balancing these competing losses requires meticulous adjustment of the tradeoff parameter $\gamma$, a process that is both challenging and time-consuming. In contrast, our MLO-MAE method tackles this challenge through a sequential process integrated into an end-to-end framework. Initially, the method involves pretraining the encoder. Following this, the pretrained encoder is transitioned to the next stage, where the classification head is trained. The

Table 7: Linear probing accuracy (%) on CIFAR-10 under different unrolling steps.

| Unrolling steps | 1 | 2 | 5 |
|---|---|---|---|
| Accuracy (%) | 83.8 | 84.3 | **84.6** |

Table 8: Linear probing accuracy (%) on CIFAR-10 under different patch sizes.

| Patch size | 2 | 4 | 8 |
|---|---|---|---|
| Accuracy (%) | **84.3** | 82.1 | 81.9 |

Table 9: Mean intersection over Union (MIoU) in using semantic segmentation as the downstream task on ADE20K

| Method (mIoU) | **MAE** | **MLO-MAE** (Ours) |
|---|---|---|
| ADE20K | 28.8 | 29.1 |

pretraining task in MLO-MAE aids the classification task by providing an effective image encoder, instead of competing with the classification task.

**Unrolling steps.** In this study, we explored how the number of unrolling steps in the Optimization Algorithm, as detailed in Section 3.2, affects the final performance. The study was performed on CIFAR-10, with linear probing as the evaluation protocol. Table 7 shows linear probing accuracy under different unrolling steps. The results reveal that an increase in the unrolling steps leads to a gradual improvement in accuracy. This enhancement can be attributed to the fact that a higher number of unrolling steps allows for more frequent updates to the image encoder weights (with more iterations in Stage I), prior to any updates being made to the classification head and masking network. Consequently, this yields a more refined gradient estimation for the parameters of the classification head and masking network, as these estimations are based on the image encoder that has undergone more extensive training. However, it is important to note that increasing the number of unrolling steps also brings in a notable computational overhead. In this context, our observations indicate that while increasing the unrolling step from one to two leads to a 0.5% boost in CIFAR-10 linear probing performance, the gain diminishes to just 0.3% when the unrolling steps are further raised from two to five. This suggests a diminishing return in performance improvement relative to the increased computational demand.

**Masking ratios.** We studied how the masking ratio during pretraining affects downstream task performance, using the CIFAR-10 dataset and a linear probing protocol. The results, illustrated in Figure 3, reveal that intermediate masking ratios deliver optimal performance. When the masking ratio is too low, the reconstruction task becomes overly simple, failing to push the image encoder to develop robust representations. Conversely, an excessively high masking ratio makes the task overly challenging, which also impedes the learning of effective representations. Notably, our method demonstrates resilience to changes in masking ratios ranging from 60% to 80%. Within this interval, there is minimal fluctuation in the results of linear probing. Additional experiment on dynamic mask ratio can be found in Appendix D.4.

**Patch sizes.** In this study, we investigated the impact of different image patch sizes on the performance of our method when applied to the CIFAR-10 dataset, utilizing the linear probing evaluation protocol. Our experiments focused on three patch sizes: $2 \times 2$, $4 \times 4$, and $8 \times 8$. The results, presented in Table 8, show that the $2 \times 2$ patch size achieves a linear probing accuracy of 84.3%, which is 2.4% (absolute) higher than that obtained with the larger $8 \times 8$ patch size. These findings suggest that MLO-MAE is more effective when employing a larger number of smaller patches, particularly for small images, such as those in the CIFAR-10 dataset with a size of $32 \times 32$. The reason is that smaller patches allow for more precise and detailed candidate masks. This leads to better feature representation learning. Moreover, smaller patches provide a finer grid over the image, allowing the model to capture more detailed and subtle features. This is particularly beneficial for small images, where each pixel can carry significant information.

**Semantic segmentation as downstream feedback.** In the base MLO-MAE framework, image classification was used as the downstream task to guide the masking network. In this section, we evaluate the generalizability of MLO-MAE by adapting semantic segmentation as the downstream task in Stages II and III. Specifically, we adopted the ADE20K dataset for both pretraining and semantic segmentation evaluation. The results, summarized in Table 9, demonstrate that when semantic segmentation is used as the downstream feedback, MLO-MAE consistently outperforms the baseline. This highlights that the effectiveness of MLO-MAE is not limited to a specific downstream task, underscoring its versatile and robust design.

Table 10: Pretraining time (GPU hours measured on A100).

|  | **MAE** | **SemMAE** | **AutoMAE** | **MLO-MAE** (Ours) |
|---|---|---|---|---|
| ImageNet-1K | 2132 hrs | 1154 hrs | 1344 hrs | 1083 hrs |

## 4.6 Computational costs

Although MLO-MAE introduces additional computational overhead due to its multi-level optimization, this is balanced by its lower epoch requirement for achieving convergence. In contrast to the 800 epochs needed for standard MAE and SemMAE, MLO-MAE converges in just 50 epochs. This significant reduction in the number of epochs effectively offsets the increased computational demands. The total GPU hours (on Nvidia A100 GPU) for MLO-MAE on ImageNet, as shown in Table 10, amount to 1083. This number is less than those of baselines while our method achieves better test accuracy than baselines as shown in Tables 1 and 2.

## 5 Conclusion

In this paper, we proposed MLO-MAE, a method that automatically learns an optimal masking strategy in Masked Autoencoder (MAE) by leveraging feedback from downstream tasks. Unlike the vanilla MAE which applies uniform patch masking irrespective of their informativeness, MLO-MAE adaptively concentrates on more informative image regions. Different from other MAE methods that do not utilize feedback from downstream tasks for masking-strategy optimization, MLO-MAE uniquely capitalizes on such feedback to refine its masking approach. Our experiments across various datasets demonstrate that MLO-MAE outperforms MAE baselines by learning downstream task guided masking strategies. Furthermore, the representations generated by MLO-MAE exhibit high transferability to a range of downstream tasks, including fine-grained classification, semantic segmentation, and object detection, highlighting our method's versatility and effectiveness.

## 6 Broader impact

Our paper presents the Multi-level Optimization for Masked Autoencoders (MLO-MAE) framework, enhancing self-supervised learning in visual data processing. MLO-MAE can have a broad impact across various fields like medical imaging, autonomous vehicles, and content moderation. In the healthcare sector, it could improve disease detection and diagnosis. Similarly, in the realm of autonomous driving, it may enhance object recognition for safer vehicle automation. However, we also recognize the ethical considerations and potential risks associated with the application of our work. The increased capability of image processing models can lead to concerns around privacy, surveillance, and the potential misuse of technology in unauthorized or harmful ways. The deployment of such technologies must be guided by ethical principles and regulatory frameworks to protect individual privacy and prevent misuse.

## 7 Limitations

Our multi-level optimization framework introduces two main limitations. First, the hypergradient computation incurs significant computational overhead. We mitigate this by using gradient approximation, where we control the approximation level through the number of unrolling steps. This trade-off balances computational cost with optimization accuracy. Second, our approach increases GPU memory usage due to the inclusion of a classification head and the need to load three separate training datasets simultaneously. Although our current implementation does not explicitly address this, techniques like gradient accumulation could reduce memory demands. Future work can focus on developing more efficient multi-level optimization algorithms to systematically overcome these limitations.

## Acknowledgments

This research was supported by NSF IIS2405974 and NSF IIS2339216.

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

# A    Optimization Algorithm

We develop an efficient optimization algorithm to solve the MLO-MAE problem demonstrated in Figure 1. Notations are given in Table 11.

Table 11: Descriptions of notations used in optimization algorithm in Appendix A.

| Notation | Description |
|---|---|
| $E$ | Backbone encoder that takes in input patches and generates representation embedding |
| $D$ | Backbone decoder that takes in representation embedding and generates the reconstructed image |
| $C$ | Classification head |
| $T$ | Masking network |
| $\mathcal{D}$ | Image classification dataset |
| $\mathcal{D}^{tr}$ | Train set split based on $\mathcal{D}$ |
| $\mathcal{D}^{val}$ | Validation set split based on $\mathcal{D}$ |
| $\mathcal{D}_u$ | Unlabeled dataset based on $\mathcal{D}$ |
| $X$ | An arbitrary image from $\mathcal{D}$ |
| $\mathcal{M}(.)$ | Masked image |
| $\mathcal{L}_{rec}(.)$ | MAE image reconstruction loss |
| $\mathcal{L}_{cls}(.)$ | Cross entropy image classification loss |
| $\sigma(.)$ | Sigmoid activation function that produces masking probability given our masking network |
| $\eta_E$ | Learning rate for updating the encoder |
| $\eta_C$ | Learning rate for updating the classification head |
| $\eta_T$ | Learning rate for updating the the masking network |
| $r$ | Masking ratio |

## A.1    Implicit Differentiation for Gradient Computation

In the MLO-MAE framework, we adopt implicit differentiation as a key tool for computing gradients in scenarios characterized by complex, nested optimization structures. This approach is particularly effective when dealing with variables implicitly interconnected. To solve the MLO-MAE optimization problem, we utilize a robust algorithm first introduced in (Choe et al., 2022). This algorithm is underpinned by a solid theoretical framework, and its convergence properties have been extensively examined in recent scholarly contributions. Each stage of the optimization process necessitates the identification of an optimal solution, denoted with an asterisk (*) and positioned on the left-hand side of the equation. Computing this exact optimal solution is typically resource-intensive. To manage this efficiently, we apply the strategy proposed in (Liu et al., 2018), which involves approximating the optimal solution via a one-step gradient descent update. This approximation is then integrated into the next level of the optimization process. In our analysis, the symbol $\frac{\partial \cdot}{\partial \cdot}$ is used to represent partial derivatives, while $\frac{d \cdot}{d \cdot}$ signifies ordinary derivatives. The term $\nabla^2 f(X, Y)$ denotes the second-order partial derivative of $f(X, Y)$ with respect to $Y$ and $X$, formalized as $\frac{\partial^2 f(X,Y)}{\partial X \partial Y}$. The initial phase of our methodology involves approximating $E^*(T)$ as follows:

$$E^*(T) \approx E' = E - \eta_E \cdot \nabla_E \sum_{X \in \mathcal{D}_u} \sum_{j=1}^{N \times r} \sigma(\mathcal{M}_j(X; T, r), X; T) \mathcal{L}_{rec}(X - \mathcal{M}(X; T, r), \mathcal{M}_j(X; T, r); E, D) \quad (6)$$

where $\eta_E$ denotes the learning rate. Subsequently, $E'$ is substituted into $\mathcal{L}_{cls}(E'(T), C, \mathcal{D}^{tr})$ to yield an approximated objective function. Similarly, $C^*(E^*(T))$ is approximated using a single-step gradient descent with respect to this approximated objective:

$$C^*(E^*(T)) \approx C' = C - \eta_C \cdot \nabla_C \mathcal{L}_{cls}(\mathcal{D}^{tr}; E'(T), C) \tag{7}$$

Finally, $E'(T)$ and $C'(E'(T))$ are incorporated into $\mathcal{L}_{cls}(E'(T), C'(E'(T)), \mathcal{D}^{val})$ to obtain an approximated version of the objective function. The parameter $T$ is then updated using gradient descent:

$$T \leftarrow T - \eta_T \cdot \nabla_T \mathcal{L}_{cls}(\mathcal{D}^{val}; E'(T), C'(E'(T))) \tag{8}$$

By applying the chain rule to this approximation, we obtain:

$$\nabla_T \mathcal{L}_{cls}(\mathcal{D}^{val}; E'(T), C'(E'(T))) = \frac{\partial \mathcal{L}_{cls}^{val}}{\partial T} + \frac{\partial \mathcal{L}_{cls}^{val}}{\partial E'} \frac{\partial E'}{\partial T} + \frac{\partial \mathcal{L}_{cls}^{val}}{\partial C'} \frac{\partial C'}{\partial E'} \frac{\partial E'}{\partial T} \tag{9}$$

where:

$$\frac{\partial E'}{\partial T} = -\eta_E \cdot \nabla_{E,T}^2 \sum_{X \in \mathcal{D}_u} \sum_{j=1}^{N \times r} \sigma(\mathcal{M}_j(X; T, r), X; T) \mathcal{L}_{rec}(X - \mathcal{M}(X; T, r), \mathcal{M}_j(X; T, r); E, D) \tag{10}$$

and

$$\frac{\partial C'}{\partial E'} = \eta_C \cdot \nabla_{C,E'}^2 \mathcal{L}_{cls}^{tr} \tag{11}$$

Here, $\mathcal{L}_{cls}^{val} = \mathcal{L}_{cls}(\mathcal{D}^{val}; E'(T), C'(E'(T)))$ and $\mathcal{L}_{cls}^{tr} = \mathcal{L}_{cls}(\mathcal{D}^{tr}; E'(T), C)$.

## A.2 Finite Difference Approximation for Gradient Estimation

To reduce the complexity of solving MLO-MAE, we utilize the Finite Difference Approximation (FDA) method, particularly in estimating gradients where analytical differentiation is challenging. Specifically, directly computing Jacobian vector multiplication with MLO problems is computationally expensive, which can be efficiently approximated by FDA methods (Choe et al., 2022).

FDA approximates the gradient of a function by computing the change in the function value for a small perturbation in the input. For example, for a function $f(x)$, the gradient approximation is given by:

$$\nabla f(x) \approx \frac{f(x + \delta x) - f(x)}{\delta x} \tag{12}$$

In our experimental framework, we extended the formula above to compute hypergradients in MLO problems. Specifically, we incorporated the bi-directional finite difference approximation (FDA) method for gradient estimation within complex, nested optimization contexts. This technique is particularly pertinent in scenarios where traditional analytical gradient computation is either impractical or excessively resource-intensive.

The bi-directional FDA extends the conventional finite difference approach by introducing perturbations in both positive and negative directions relative to the current parameter values. This methodology provides a more nuanced and accurate gradient estimation compared to one-sided finite difference methods.

Each parameter in our current optimization problem undergoes an initial positive perturbation, determined by a predefined small step size $\epsilon$. Post this perturbation, we compute the loss and the corresponding gradients with respect to the preceding optimization problem's parameters. A subsequent negative perturbation, amounting to double the initial epsilon, shifts the parameters below their original values for a re-evaluation of the loss and re-calculation of gradients.

In our Multi-level Optimization (MLO) framework, the bi-directional FDA offers substantial benefits. It enables efficient gradient estimation in situations where conventional backpropagation may fail, especially in

handling the complex, implicit dependencies between parameters. This approach is instrumental in enhancing our optimization techniques within intricate MLO settings.

Our implementation of the bi-directional FDA is finely tuned to strike a balance between computational efficiency and the accuracy of gradient estimation. The selection of the epsilon value is critical, aiming to minimize numerical instability while maintaining sensitivity to changes in parameters. This balance is essential for ensuring the robustness of the gradient estimation process in the stochastic realm of machine learning models.

### A.3 Integration of Methods

Implicit differentiation and finite difference approximation are integrated to balance theoretical accuracy with computational feasibility. This combination enhances the robustness and efficiency of our optimization process in the MLO-MAE framework. In the application of these optimization methods, several key considerations are taken into account. Firstly, the choice of $\Delta T$ in the finite difference approximation is a critical factor, as it directly influences the accuracy and stability of the gradient estimation. An appropriate value for $\Delta T$ ensures a balance between precision and numerical stability. Secondly, we address the computational complexity problem inherent in implicit differentiation. This aspect is particularly relevant in deep network architectures, where computational resources can be a limiting factor. To mitigate this, we optimize the use of implicit differentiation to balance computational demands with the need for accurate gradient computation. Lastly, maintaining numerical stability is paramount in both methods. Techniques such as gradient normalization and careful arithmetic handling are employed to ensure that the computations remain stable, especially in scenarios where small numerical errors can significantly impact the overall results. This comprehensive optimization algorithm is pivotal in enabling the efficient training and validation of the MLO-MAE framework.

---

**Algorithm 1** MLO-MAE Optimization Algorithm

---

1: **Input:** Training dataset $\mathcal{D}_{tr}$, validation dataset $\mathcal{D}_{val}$
2: **Output:** Optimized parameters $E^*(T)$, $D^*(T)$, $C^*(E^*(T))$, $T^*$
3: **procedure** MAE PRETRAINING($\mathcal{D}_u, T$)
4:     Initialize encoder $E$ and decoder $D$
5:     **for** each image $X_i \in \mathcal{D}_u$ **do**
6:         Patchify image into $\bar{X}_i$
7:         Compute masked image using $\sigma(\mathcal{M}_j(X;T,r), X;T)$
8:         Update $E$, $D$ by minimizing $\sum_{X \in \mathcal{D}_u} \sum_{j=1}^{N \times r} \sigma(\mathcal{M}_j(X;T,r), X;T)\mathcal{L}_{rec}(X - \mathcal{M}(X;T,r), \mathcal{M}_j(X;T,r); E, D)$
9:     **end for**
10:     **return** $E^*(T)$, $D^*(T)$
11: **end procedure**
12: **procedure** CLASSIFICATION HEAD TRAINING($E^*(T), \mathcal{D}_{tr}$)
13:     Freeze $E^*(T)$, initialize classifier $C$
14:     **for** each image $X_i \in \mathcal{D}_{tr}$ **do**
15:         Update $C$ by minimizing $\mathcal{L}_{cls}(\mathcal{D}_{tr}; E^*(T), C)$
16:     **end for**
17:     **return** $C^*(E^*(T))$
18: **end procedure**
19: **procedure** VALIDATION OPTIMIZATION($E^*(T), C^*(E^*(T)), \mathcal{D}_{val}$)
20:     Freeze $E^*(T)$, $C^*(E^*(T))$
21:     Optimize $T$ by minimizing $\mathcal{L}_{cls}(\mathcal{D}_{val}; E^*(T), C^*(E^*(T)))$
22:     **return** $T^*$
23: **end procedure**

---

## A.4 Differentiability of MLO-MAE Framework

In the initial phase of our Multi-level Optimization Masked Autoencoder (MLO-MAE) framework, the focus is on the reconstruction loss and its differentiability relative to the parameters of the learnable masking network. This network is pivotal, as it determines the masking patterns for the input data, directly impacting the autoencoder's reconstruction loss. The key to effective gradient-based optimization lies in ensuring the differentiability of this reconstruction loss with respect to the masking network's parameters. Due to the inherently discrete nature of mask selection, integrating the masking network directly into the MAE's reconstruction loss initially leads to non-differentiability issues. To circumvent this, our approach employs a sigmoid activation function, denoted as $\sigma(.)$, to generate soft masks. These soft masks assign a continuous value between 0 and 1 to each image patch, indicating the likelihood of that patch being masked. This likelihood is learned from both masked and unmasked patches. In this first phase, we tackle non-differentiability by utilizing the MAE reconstruction loss together with the masking probability. This is achieved as illustrated in Equation (1), which allows the proportional contribution of each patch to influence the overall loss, facilitating differentiation with respect to the network parameters through the application of $\sigma(\cdot)$.

To further address the non-differentiability issue in our MLO-MAE framework incurred by mask selection, the SoftSort (Prillo & Eisenschlos, 2020) technique—a differentiable approximation of sorting operations—replacing the conventional, non-differentiable "argsort" operation, can be integrated. SoftSort enables the learning of a continuous relaxation of sorting, vital for gradient-based optimization and for crafting nuanced, performance-enhancing masks. This represents a significant leap in refining the effectiveness of the reconstruction process. However, given the computational demands of SoftSort, we propose a simplified approach that approximates the base Jacobian with an identity matrix similar to (Choe et al., 2023), thereby simplifying the gradient during backpropagation and enhancing the efficiency by "jumping over" the non-differentiable argsort operation. By treating non-differentiable operations as identity functions during backpropagation, this method allows gradients from the reconstruction loss to flow as if the masking operations were inherently differentiable. This strategy significantly speeds up training by enabling the use of standard gradient-based optimization techniques. In our MLO-MAE, the masking network employs soft masks, which, through a sigmoid activation function $\sigma(.)$, assign each image patch a continuous value from 0 to 1. This assignment reflects the probability of masking, informed by both masked and unmasked patches. By approximating the gradients for non-differentiable functions as identity functions during backpropagation, our method enables the differentiation of the MAE reconstruction loss, $\mathcal{L}_{rec}$, relative to the masking network parameters $T$. This differentiation is facilitated by incorporating the output probabilities of $\sigma(.)$ into the reconstruction loss $\mathcal{L}_{rec}$, as detailed in Equation (1).

## A.5 Convergence Property of the MLO-MAE framework

Previous research has extensively analyzed the convergence properties of optimization algorithm for bi-level optimization (BLO) problems. For instance, Ji et al. (2021) establish sharper non-asymptotic convergence rates for implicit differentiation (AID) and iterative differentiation (ITD) methods, addressing both deterministic and stochastic bilevel problems with improved computational complexity. Grazzi et al. (2020) analyze the iteration complexity of hypergradient computation, providing explicit non-asymptotic convergence bounds and demonstrating the efficiency of AID with conjugate gradient under contraction assumptions. Bao et al. (2021) focus on the stability and convergence of bilevel optimization by deriving expectation bounds for the generalization gap, emphasizing how regularization strategies improve convergence. Bao et al. (2021) address bilevel problems with nonconvex lower-level structures, introducing the IAPTT-GM framework, which ensures convergence even in the absence of lower-level convexity. Zhang et al. (2021) propose a convergence-guaranteed approach for bilevel optimization using stochastic implicit gradients, ensuring stationary point convergence while reducing computational overhead.

While the analyses in the aforementioned works are primarily on bi-level optimization problems, the convergence proof can be readily extended to the multi-level cases. In addition our emipirical results demonstrate that MLO-MAE framework consistently converges and outperforms the baseline methods as shown in Section 4. The training loss curves in Fig 4 also corroborate with our empirical convergence finding.

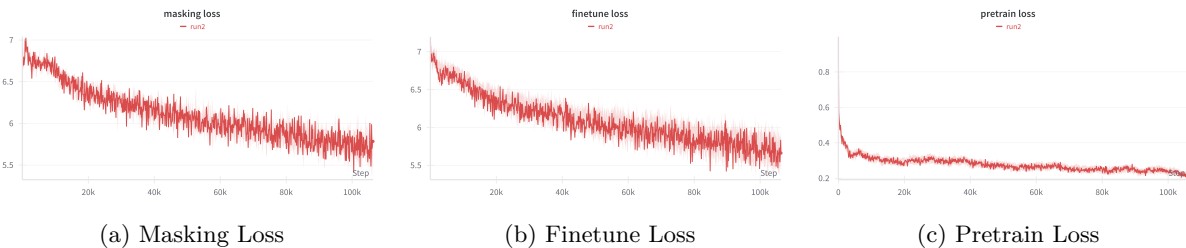

(a) Masking Loss     (b) Finetune Loss     (c) Pretrain Loss

Figure 4: Loss curves for masking, fine-tuning, and pretraining.

# B    Detailed Experimental Settings

## B.1    Datasets

**CIFAR-10 (Krizhevsky et al., a):**   CIFAR-10 is a fundamental dataset for image classification, comprising 60,000 32x32 color images across 10 classes, with 6,000 images per class. It is widely used in machine learning research because of its manageable size and diversity of images. CIFAR-10 tests the ability of our MLO-MAE framework to capture essential features in small-scale images and generalize across a variety of everyday objects.

**CIFAR-100 (Krizhevsky et al., b):**   CIFAR-100 is similar to CIFAR-10 in image size and total number of images but is significantly more challenging due to its 100 classes, each containing 600 images. The increased number of classes in CIFAR-100 allows us to evaluate the capability of MLO-MAE in a more granular classification context, providing insights into how well the model differentiates between a larger number of categories with fewer examples per category.

**ImageNet-1K (Deng et al., 2009):**   ImageNet, a subset of the larger ImageNet database, is one of the most influential datasets in the field of image classification. It contains 1.3M training images and 50K validation images categorized into 1,000 classes. The dataset's extensive size and diversity present a rigorous test for any machine learning model. Our use of ImageNet is aimed at assessing the scalability and robustness of the MLO-MAE framework in handling complex, large-scale classification tasks.

**CUB-200-2011 (Wah et al., 2011):**   The CUB-200-2011 dataset is a specialized collection designed for fine-grained bird species classification, developed by the California Institute of Technology. It contains 11,788 images, representing 200 bird species, with a focus on North American birds. Each image in the dataset is accompanied by detailed annotations, including bounding boxes, part locations, and attribute labels, making it ideal for detailed image analysis tasks. The dataset is widely used in computer vision research, particularly in tasks that require distinguishing between visually similar sub-categories.

**Stanford Cars (Krause et al., 2013):**   The Stanford Cars dataset is a large collection of car images created by researchers at Stanford University, it contains 16,185 images of 196 classes of cars. The data is split into 8,144 training images and 8,041 testing images, where each class has been split roughly in a 50-50 split. Classes are typically at the level of Make, Model, Year, ex. 2012 Tesla Model S or 2012 BMW M3 coupe.

**iNaturalist 2019 (Van Horn et al., 2018):**   The iNaturalist 2019 dataset is a comprehensive biodiversity collection used for machine learning and research, created by the iNaturalist project, a collaboration between the California Academy of Sciences and the National Geographic Society. It features over 859,000 high-quality images of 1,010 species, each with detailed metadata including species name, observation location, date, and time. This extensive dataset is crucial for species identification and, distribution modeling, and is utilized in the annual iNaturalist Challenge to enhance automated species recognition technologies.

**ADE20K (Zhou et al., 2017):** ADE20K is a comprehensive dataset for semantic segmentation, containing more than 20,000 images annotated for a variety of scenes and objects, making it one of the most diverse datasets available for this task. Each image in ADE20K is annotated with pixel-level segmentation masks, encompassing a wide range of objects and scene categories. The complexity and richness of ADE20K make it an ideal choice for testing the efficacy of the MLO-MAE framework in understanding and segmenting complex visual scenes. The dataset challenges the model to not only recognize a diverse array of objects but also understand their spatial relationships and boundaries within various contexts.

**MS-COCO (Lin et al., 2014):** MS-COCO is a comprehensive large-scale dataset used for tasks such as object detection, segmentation, keypoint detection, and image captioning, containing a total of 328,000 images. Initially released in 2014, the dataset was split into 83,000 training images, 41,000 validation images, and 41,000 test images. In 2015, an expanded test set was introduced, adding 40,000 new images to the existing test set for a total of 81,000 test images. Responding to feedback from the research community, the 2017 version adjusted the training/validation split to 118,000 training images and 5,000 validation images, while maintaining the same images and annotations as previous versions. The 2017 test set consists of 41,000 images, a subset of the 2015 test set, and the release also includes a new unannotated dataset of 123,000 images.

**PASCAL VOC 2007 (Everingham et al.):** The PASCAL VOC 2007 dataset is a prominent benchmark in the field of object detection and image segmentation. It consists of 9,963 images annotated with 24,640 objects across 20 distinct classes, including everyday items like cars, cats, and chairs. The dataset provides comprehensive bounding box annotations for each object, facilitating the training and evaluation of object detection models. Additionally, it includes segmentation annotations, although they are more limited compared to later versions. The images in PASCAL VOC 2007 vary in size and aspect ratio, reflecting diverse real-world conditions. This dataset is divided into training, validation, and test sets, with the test set publicly available, making it an invaluable resource for benchmarking and comparing the performance of different object detection algorithms. Its widespread adoption in the research community has contributed to significant advancements in computer vision, serving as a foundational dataset for evaluating the effectiveness of new detection methods and models.

### B.2 Masking Network

To facilitate the mask generation process, we design a lightweight masking network with two linear layers, one ReLU layer in between, and one sigmoid activation. The first linear layer can be expressed in PyTorch code as nn.Linear(num_patches × emb_dim, 512), where num_patches is the number of patches from the original image (e.g. ImageNet sample of size $224 \times 224$ with patch size of $16 \times 16$ will generate $14 \times 14 = 196$ patches) and emb_dim is the embedding dimension from ViT-B (768 in our case). We use a hidden size of 512 for the output dimension of the first and the second linear layer. The ReLU layer is expressed as nn.ReLU(). The second linear layer can be expressed as nn.Linear(512, num_patches) where the output is the masking probability for each image patch with corresponding order. We pass the resulting tensor from the second linear layer through the sigmoid activation to generate values between 0 and 1.

### B.3 ImageNet

We adopt the default ViT-B model that has been employed in the original MAE. In total, we train 50 epochs for all three stages. For ImageNet experiments, we use an image size of $224 \times 224$.

**Pretraining** Detailed three-stage MLO-MAE pretraining setting is in Table 12. We follow MAE settings on data augmentations by only using RandomResizedCrop and RandomHorizontalFlip. For MLO-MAE in Table 1 experiment, we train MLO-MAE using xavier_uniform (Glorot & Bengio, 2010). We follow the linear lr scaling as used in the MAE. We randomly split the training set of ImageNet by a ratio of 80/20 to be the new training set and the new validation set. We use the same new training set in Stage I and Stage II for training, while use the new validation set in Stage III for training the masking network. We use the original ImageNet validation set to report validation accuracy in Stage II.

Table 12: MLO-MAE ImageNet Pretraining Settings.

| MLO-MAE Stage | Config | value |
|---|---|---|
| Stage I | model | ViT-B |
| | optimizer | AdamW |
| | base learning rate | $1e-4$ |
| | weight decay | 0.05 |
| | image size | 224 |
| | patch size | $16 \times 16$ |
| | optimizer momentum | $\beta_1, \beta_2 = 0.9, 0.95$ |
| | batch size | 256 |
| | learning rate scheduler | cosine anneal |
| | unrolling steps | 1 |
| | mask ratio | 0.75 |
| Stage II | learning rate | $4e-5$ |
| | weight decay | 0.05 |
| | image size | 224 |
| | optimizer momentum | $\beta_1, \beta_2 = 0.9, 0.95$ |
| | batch size | 256 |
| | learning rate scheduler | cosine anneal |
| | unrolling steps | 1 |
| Stage III | learning rate | $4e-5$ |
| | masking network | nn.Linear(num_patches $\times$ emb_dim, 512) nn.ReLu nn.Linear(512, num_patches) torch.sigmoid() |
| | weight decay | 0.05 |
| | image size | 224 |
| | optimizer momentum | $\beta_1, \beta_2 = 0.9, 0.95$ |
| | batch size | 256 |
| | learning rate scheduler | cosine anneal |
| | unrolling steps | 1 |

**Linear Probing**  Due to the inherent design of our three-level optimization framework, we do not conduct separate linear probing and directly report the Stage II testing accuracy (test dataset not seen in MLO-MAE training). Therefore, we report the training setting as in Stage II shown in Table 12.

**Fine-tuning**  We directly followed the MAE fine-tuning experiments and did not make additional changes. Detailed settings can be found in Table 9 of MAE (He et al., 2022).

## B.4  CIFAR

We adopt the same ViT-B architecture as in B.3 but change the input image size and patch size to be 32 and 2 respectively. In total, we train MLO-MAE 50 epochs for all three stages.

**Pretraining**  We pretrain our model from scratch (i.e. no pretrained initialization from other datasets) using the MLO-MAE method on two CIFAR datasets. Table 13 shows the detailed training setting for CIFAR-10 and CIFAR-100 experiments. We adopted a similar setting from the ImageNet experiment, with

Table 13: MLO-MAE CIFAR-10/100 Pretraining Settings.

| MLO-MAE Stage | Config | value |
|---|---|---|
| Stage I | model | ViT-B |
| | optimizer | AdamW |
| | base learning rate | $1e-3$ |
| | weight decay | 0.05 |
| | image size | 32 |
| | patch size | $2 \times 2$ |
| | optimizer momentum | $\beta_1, \beta_2 = 0.9, 0.95$ |
| | batch size | 128 |
| | learning rate scheduler | cosine anneal |
| | unrolling steps | 2 |
| | mask ratio | 0.75 |
| Stage II | learning rate | $1e-3$ |
| | weight decay | 0.05 |
| | image size | 32 |
| | optimizer momentum | $\beta_1, \beta_2 = 0.9, 0.95$ |
| | batch size | 128 |
| | learning rate scheduler | cosine anneal |
| | unrolling steps | 1 |
| Stage III | learning rate | $1e-3$ |
| | masking network | nn.Linear(num_patches $\times$ emb_dim, 512) nn.ReLu nn.Linear(512, num_patches) torch.sigmoid() |
| | weight decay | 0.05 |
| | image size | 32 |
| | optimizer momentum | $\beta_1, \beta_2 = 0.9, 0.95$ |
| | batch size | 128 |
| | learning rate scheduler | cosine anneal |
| | unrolling steps | 1 |

minor modifications on learning rate, data augmentation, image size, and patch size. We use conventional RandomCrop and RandomHorizontalFlip on both CIFAR-10 and CIFAR-100. We pretrain with MLO-MAE for 200 epochs.

**Linear Probing** Similar to section B.3, we report the training setting as in Stage II shown in Table 13.

**Fine-tuning** For CIFAR fine-tuning, we use lr=$1e-4$, weight_decay=$5e-5$, optimizer=AdamW, batch_size=64, and epoch=100. We use default data augmentation on CIFAR, including RandomCrop, Resize, and RandomHorizontalFlip. We maintain the image size to be 32. This experiment is performed on top of MLO-MAE CIFAR-10/100 pretrained weights respectively.

## B.5 Classification on Fine-grained Datasets

We perform image classification using MLO-MAE pretrained ViT-B on CUB-200-2011 (Wah et al., 2011), Stanford Cars (Krause et al., 2013), and iNaturalist 2019 (Van Horn et al., 2018) fine-grained datasets. We

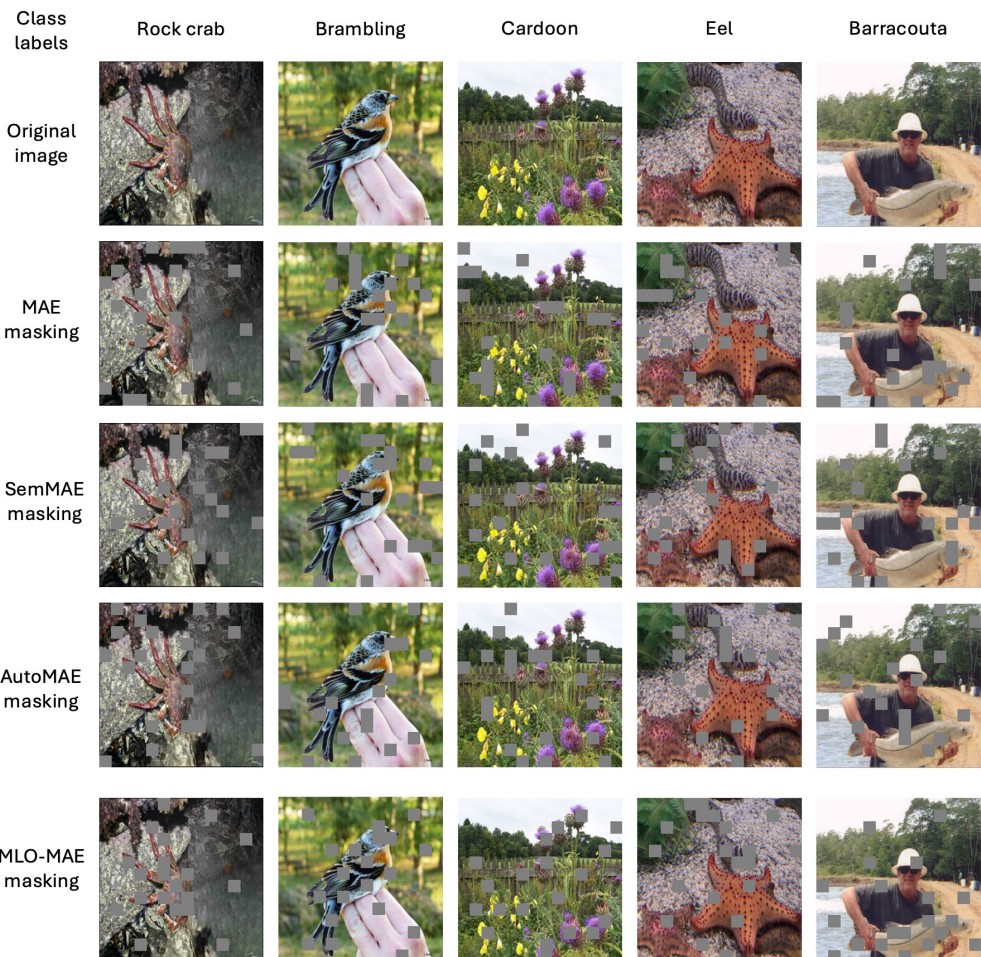

Figure 5: Visualization of the masking patterns of MLO-MAE and baselines on randomly sampled ImageNet images.

follow the setting from MAE (He et al., 2022) with minor adjustments on learning rate and epochs. These experiments are performed on top of MLO-MAE ImageNet-1K pretrained weights respectively.

### B.6  Semantic segmentation on ADE20K

We use the semantic segmentation code implementation of MAE by MMSegmentation (Contributors, 2020). Given a ViT-B model pretrained on ImageNet using MLO-MAE or a baseline and subsequently fine-tuned on ImageNet, to transfer it for semantic segmentation, we integrated it as a backbone model into the UPer-Net (Xiao et al., 2018) semantic segmentation framework. It was then further fine-tuned on the challenging ADE20K dataset (Zhou et al., 2017) containing 25K images spanning 150 semantic categories. The fine-tuning was conducted by the AdamW optimizer for over 160,000 iterations, with a batch size of 8 and a learning rate of 0.0001.

## C  Visualization

### C.1  Masking Pattern Visualization

We randomly sampled five images from ImageNet and visualized the masked patches learned by our method MLO-MAE. We also included visualizations for baseline methods, including MAE, SemMAE, and AutoMAE,

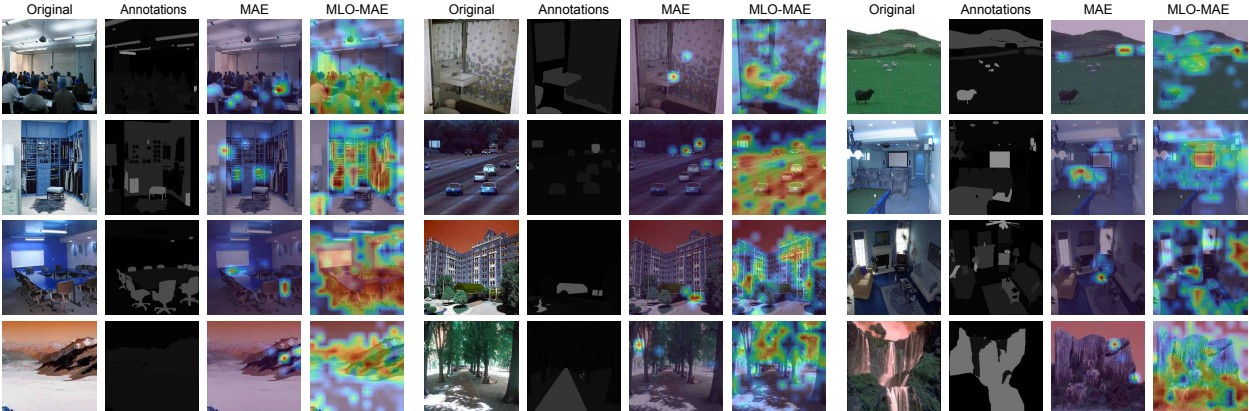

Figure 6: Visualizing 2D activations of MAE and MLO-MAE pretrained ViT-B weights on 12 examples of ADE20K validation set using GradCAM. For each set of images, from left to right, are original image, ground truth annotations, MAE activation map, and MLO-MAE activation map. For both MAE and MLO-MAE models, features are extracted from the `norm1` layer from the last ViT block. Red colors high activation region.

with a masking ratio of 10%. As shown in Figure 5, the majority of the masked patches learned by MLO-MAE are on foreground objects directly relevant to the class labels of these images. In contrast, MAE, SemMAE, and AutoMAE place the majority of masked patches on background regions irrelevant to image class labels. These results indicate that MLO-MAE encourages the encoder network to focus on learning effective representations for objects rather than background regions. By focusing on correctly reconstructing the masked patches in object regions, the encoder can effectively capture the intrinsic properties of the objects.

MLO-MAE achieves this ability by leveraging the downstream classification task to guide the pretraining of the encoder and the learning of the masking strategy. Minimizing the validation loss of the downstream classification task in Stage III of MLO-MAE encourages the pretrained encoder to learn discriminative representations that can distinguish between different classes. To learn these discriminative representations, MLO-MAE emphasizes masking and reconstructing patches in object regions, as these objects are directly related to the class labels. In contrast, SemMAE and AutoMAE use the attention maps produced by StyleGAN and adversarial learning to mask patches. These attention maps are created without leveraging guidance from the class labels of the downstream classification task, resulting in SemMAE and AutoMAE being less effective at masking class-label-relevant objects compared to MLO-MAE.

It is worth noting that MLO-MAE emphasizes masking objects directly related to the image class label rather than any objects. For instance, in the fourth image, which contains two types of objects - eel and starfish, MLO-MAE places more masked patches on the eel because the class label of the image is eel. Although starfish are prominent objects in this image, MLO-MAE masks fewer patches on it since the image's class label is not starfish. Again, this targeted masking strategy is learned with guidance from the downstream image classification task, aimed at enhancing classification accuracy.

## C.2 GradCAM visualization

To delve into the representation learning prowess of MLO-MAE, we also present a visual analysis of activation maps generated by both the pretrained MAE and MLO-MAE ViT-B models. Specifically, we examine 12 examples from the ADE20K validation set, as depicted in Figure 6. These activation maps are derived from features extracted from the `norm1` layer within the final ViT block of the backbone architecture. Our comparative analysis reveals that MLO-MAE consistently produces activation maps of notably higher semantic coherence compared to MAE. This enhancement is attributed to the tailored guidance provided by task-specific masking during the pretraining phase. Notably, MLO-MAE demonstrates a proficiency in pin-

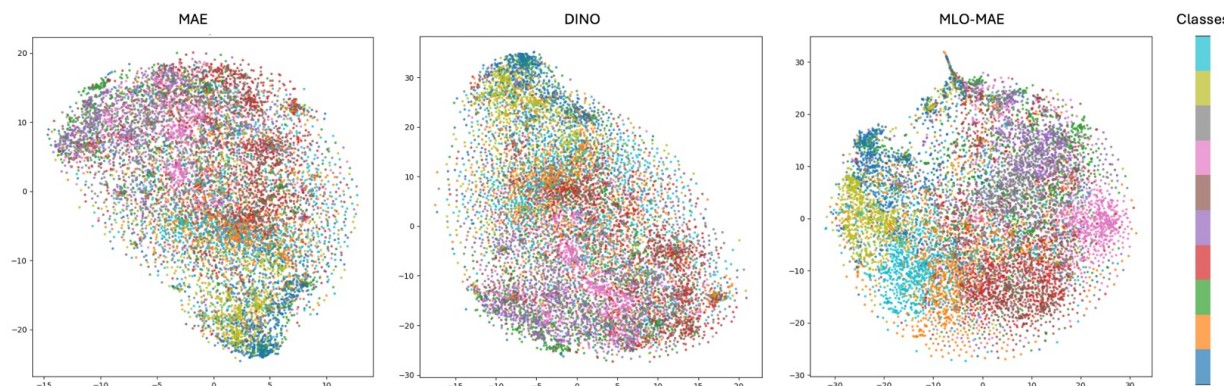

Figure 7: t-SNE visualizations of representations learned by MAE, DINO, and MLO-MAE for CIFAR-10 test images.

pointing highly informative regions, demonstrating the efficacy of the MLO-based pretraining methodology in refining visual representation learning.

### C.3 t-SNE Visualization

Table 14: Ratio of average intra-class similarity to inter-class similarity for representations extracted by encoders pretrained using MLO-MAE, MAE, and DINO on the test images of CIFAR-10, CIFAR-100, and ImageNet.

| Method | Ratio on CIFAR-10 | Ratio on CIFAR-100 | Ratio on ImageNet |
|---|---|---|---|
| MAE | 1.25 | 1.19 | 1.08 |
| DINO | 1.31 | 1.27 | 1.11 |
| MLO-MAE | **1.59** | **1.52** | **1.43** |

We utilized the encoders learned by MLO-MAE, MAE, and fully supervised method, DINO (Caron et al., 2021), to extract representations for the test images of CIFAR-10. These representations were then visualized using t-SNE, as shown in Figure 7. The visualization indicates that in the MLO-MAE representation space, different classes are better separated, with images from the same class grouped together. In contrast, the MAE and DINO representations show a mixing of different classes. Furthermore, we measured the ratio between intra-class similarity and inter-class similarity. For intra-class similarity, we calculated the cosine similarity between the representations of each pair of images within the same class and averaged these values. Likewise, for inter-class similarity, we computed the average cosine similarity for pairs of images from different classes. The results are in Table 14. MLO-MAE achieves the highest ratios across all datasets, demonstrating its learned representations can better distinguish between different classes. This can be attributed to our method's use of validation loss from the downstream classification task to guide pretraining, resulting in more discriminative representations.

## D   Additional Experiments

### D.1   Object Detection

We evaluate the transfer ability of MLO-MAE ViT-B model to object detection task on PASCAL VOC 2007 dataset. Following a similar setup as in Section 4.3, we directly train the pretrained model for the object detection task using the following procedure. The dataset is downloaded and organized into training, validation, and test sets. Images are preprocessed to a fixed size (e.g., 512x512 pixels) and augmented with

Table 15: Object detection result of MLO-MAE on PASCAL VOC 2007 detection task.

| Method | ViT-B | MAE | MLO-MAE |
|--------|-------|-----|---------|
| mAP (%) | 79.1 | 82.8 | **83.5** |

techniques such as random cropping, flipping, and normalization to enhance model robustness. The ViT-Base backbone is pretrained using MAE and MLO-MAE on a large corpus of unlabeled images to learn rich visual representations. An object detection head is then attached to the backbone, consisting of fully connected layers for predicting bounding boxes and class labels. The training process employs a combination of classification loss (cross-entropy) and bounding box regression loss (smooth L1), optimized using the AdamW optimizer with a learning rate of 1e-4 and weight decay of 1e-4. We use a batch size of 16 and train the model for 50-100 epochs, incorporating early stopping based on validation performance. During training, the backbone is initialized with MAE-pretrained weights, and forward passes are performed to extract features and make predictions. The total loss is computed and backpropagation is used to update the model weights. Periodic validation monitors performance and guides hyperparameter adjustments. Evaluation metrics include mean Average Precision (mAP), precision, and recall at different IoU thresholds, alongside inference speed. For final evaluation, the trained model predicts bounding boxes and class labels on the PASCAL VOC 2007 test set, and performance is benchmarked using the mAP metric. Results may be submitted to the PASCAL VOC evaluation server for standardized comparison with other models, demonstrating the efficacy of the MAE-pretrained ViT-Base backbone in object detection tasks. Table 15 shows the result. MLO-MAE surpasses supervised ViT-B and MAE with 4.4% and 0.7%, respectively.

## D.2  Robustness

Table 16: Robustness evaluation on ImageNet variants. We use IN-1K finetuned ViT-B and directly reported from Table 1 without further training. Results are top-1 accuracy.

| Dataset | MAE | MLO-MAE |
|---------|-----|---------|
| ImageNet-A | 35.9 | **46.2** |
| ImageNet-B | 18.4 | **46.3** |
| ImageNet-C | 51.7 | **55.5** |
| ImageNet-R | 48.3 | **55.6** |
| ImageNet-S | 34.5 | **41.8** |

We evaluate the robustness of our ViT-B models on different variants of ImageNet validation sets. Following MAE's setup, we use the fine-tuned model from Table 1 without further training and only run inferences on the ImageNet robustness variants. Table 16 shows our MLO-MAE surpasses MAE in all variants with large margin.

## D.3  Computational Efficiency

Table 17: Test accuracy on CIFAR-100 with different update frequencies.

| Update frequency | Per epoch runtime (GPU hrs) | Test accuracy on CIFAR-100 (%) |
|------------------|------------------------------|--------------------------------|
| Every iteration | 0.6 | 79.4 |
| Every 5 iterations | 0.4 | 79.1 |

In Section 4.6 of the main paper, we compared the computational cost of our method to that of baseline methods, including MAE, SemMAE, and AutoMAE. The overall runtime of our method is similar to that

Table 18: Ablation on curriculum masking ratio from 0.5 to 0.9.

| Masking ratio | Test Accuracy on CIFAR 100 |
|---|---|
| 0.75 | 79.4 |
| 0.5-0.9, linear increase | 76.5 |

Table 19: Test accuracy on CIFAR-100 with full fine-tuning in stage II.

| Method | Test Accuracy on CIFAR 100 | Runtime (days on 8 GPUs) |
|---|---|---|
| Linear probing in stage II | 79.4 | 0.7 |
| Full fine-tuning in stage II | 79.5 | 1.1 |

of the baselines. Our method converges in fewer epochs but has a higher per-epoch runtime compared to the baselines. To reduce the per-epoch training time, we can decrease the update frequency of the masking network; instead of updating it every iteration (mini-batch), we update it every few iterations (e.g., every five iterations), while still updating the encoder and linear head at each iteration. Calculating the hypergradient of the masking network requires computing Jacobian matrices and performing their multiplication with vectors, which is more computationally intensive than calculating the gradient of the encoder and linear head. By reducing the update frequency of the masking network, we can significantly lower the overall computational costs. We experimented with this approach on CIFAR-100, parallelized across 8 GPUs as shown in Table 17. As can be seen, the per epoch training time is significantly reduced. Meanwhile, we empirically found that reducing the update frequency of the masking network does not significantly impact classification accuracy. This is likely because once an intermediate masking strategy is learned, it can be used for a while to pretrain the encoder without needing frequent updates.

## D.4 Curriculum Mask Ratio

Experiments performed in section 4 use fixed masking ratio, following the baseline setup. To further explore the impact of different masking behavior (fixed and dynamic masking ratio) on MLO-MAE, we experimented with a curriculum masking ratio setting. We dynamically increased the masking ratio of our method from 0.5 to 0.9 as training progressed, with a linear schedule. The results on CIFAR-100, shown in the Table 18, indicate that using a dynamic ratio does not outperform a fixed ratio of 0.75.

## D.5 Full Fine-tune in Stage II

$$\min_{T} \mathcal{L}_{cls}(\mathcal{D}^{val}; F^*(E^*(T)), C^*)$$

$$s.t.\ F^*(E^*(T)) = \underset{F,C}{\arg\min}\ \mathcal{L}_{cls}(\mathcal{D}^{tr}; F, C) + \lambda\|F - E^*(T)\|_2^2$$

$$E^*(T), D^* = \underset{E,D}{\arg\min} \sum_{X \in \mathcal{D}_u} \sum_{j=1}^{N \times r} \sigma(\mathcal{M}_j(X; T, r), X; T)\mathcal{L}_{rec}(X - \mathcal{M}(X; T, r), \mathcal{M}_j(X; T, r); E, D)$$

(13)

We experimented with full fine-tuning of the encoder during Stage II, parallelized across 8 GPUs. In this setup, full fine-tuning of the pretrained encoder involves training an encoder to have a small L2 distance from. Equation 13 is the formulation for performing full fine-tuning of the pretrained encoder during Stage II in MLO-MAE. Table 19 shows the results on CIFAR-100. As can be seen, full fine-tuning the encoder during stage II does not significantly outperform linear probing but incurs much higher computational costs. Therefore, it is preferable to fix the encoder during Stage II.

## E   Further Discussion

### E.1   Utilizing Unlabeled Data

The proposed MLO-MAE framework relies on downstream feedback to guide its masking network. This dependency naturally raises the question of how to effectively utilize unlabeled data. Here, we would like to consider two scenarios: 1) using MLO-MAE in the continued pretraining stage, and 2) direct transfer learning from the MLO-MAE pretrained model. While MLO-MAE's current formulation leverages downstream task labels to guide masking during pretraining, the framework can still operate in label-scarce settings through the following mechanisms:

**MLO-MAE in the continued pretraining stage**   MLO-MAE can be initialized with a vanilla MAE pretrained on unlabeled data. Subsequent MLO-MAE continued pretraining on the labeled data can be used to further improve the performance on downstream tasks (shown in Section 4.4), demonstrating MLO-MAE's flexibility under diverse scenarios. Note that in this scenario, the continued pretraining is happening in the finetuning stage, therefore we can assume to have access to the downstream labeled data.

**Direct Transfer Learning from MLO-MAE**   When domain-specific labels are unavailable, MLO-MAE can leverage public labeled datasets (e.g., ImageNet) to learn generalizable masking strategies. First, we initialize MLO-MAE with a standard MAE pretrained on unlabeled domain-specific data. We then pretrain it on ImageNet using its labels to guide task-aware masking, and finally transfer the model to downstream tasks without requiring their labels during pretraining. Our experiments in Section 4.3.1 show that MLO-MAE pretrained on one labeled dataset (e.g., ImageNet) transfers effectively to unseen tasks/datasets without requiring their labels during pretraining. This confirms that the learned masking strategy generalizes beyond the specific downstream task used for guidance.

### E.2   Fairness in Comparing with Baselines

While the masking network $T$ is supervised by the labels, we would like to emphasize that the comparison with baseline methods presented in Section 4 remains fair with the following clarifications:

**Equivalent Label Utilization**   All baseline methods (MAE, SemMAE, etc.) require labeled data for fine-tuning or linear probing post-pretraining. In MLO-MAE, the same labeled data is used only during pretraining (Stage II and Stage III) to guide masking, not for direct encoder updates. Thus, no additional labels are introduced—comparisons remain fair (Section 4.2).

**Empirical Validation**   Transfer learning results (Tables 3, 4, 5) demonstrate that MLO-MAE's gains persist even when pretrained on one task (e.g., ImageNet classification) and evaluated on others (e.g., ADE20K segmentation). This confirms that performance improvements stem from better representation learning.

