# OpenReview forum: "Downstream Task Guided Masking Learning in Masked Autoencoders Using Multi-Level Optimization"
_TMLR — Accepted by TMLR_

### Review · Reviewer_L73A · 2025-01-09

**Summary Of Contributions:**

This paper presents MLO-MAE, an end-to-end method for learning an optimal masking strategy for Masked Autoencoder (MAE) that incorporates feedback from downstream tasks/datasets. The training process consists of three stages, during which the masking network is optimized using Multi-level Optimization (MLO). While MLO introduces higher computational costs, the authors claim that MLO-MAE converges more quickly than baseline methods. Although the core ideas of the optimization algorithm are well-presented, I am not an expert in optimization and cannot fully guarantee its correctness.

Experimental results demonstrate that incorporating feedback from downstream datasets can enhance performance by producing more effective masking networks. Additionally, the experiments highlight MLO-MAE's ability to generalize to other datasets within the same task, transfer between different tasks, and perform in a continued pretraining setting. The ablation studies provide valuable insights into the effectiveness of three-level optimization, the diminishing returns of increasing unrolling steps, the selection of appropriate masking ratios, and the superior performance achieved with smaller patch sizes.

**Audience:**

Yes

**Broader Impact Concerns:**

No other broader impact concern.

**Claims And Evidence:**

Yes

**Requested Changes:**

Overall, I find this paper well-written and easy to follow. The following items are suggestions/questions that would simply strengthen the work:

- In Section 3.5.2, the results demonstrate the capability of transferring from classification to semantic segmentation. I wonder how the method would perform if trained solely using the feedback from semantic segmentation.
- In Section 3.6, it would be helpful to include results for MLO-MAE's performance without pretraining on MAE.
- In Section 3.8, could the authors provide an explanation for why MLO-MAE converges in fewer epochs compared to baseline methods?

**Strengths And Weaknesses:**

Strengths:
- This paper is well-written, with clear expression, well-presented formulations and well-organized figures/tables.
- The motivation to incorporate feedback from downstream tasks/datasets is well-justified.
- The experimental design is thoughtfully executed, providing evidence to support the advantages of the proposed MLO-MAE.

Weakness:
- Some experimental details are insufficiently explained, as outlined in the "Requested Changes" section below.

---

> ### Author Response · Authors · 2025-01-16
>
> ## 1. Feedback from semantic segmentation
> We appreciate the reviewer’s suggestion. Following the reviewer’s suggestion, we evaluated MLO-MAE on semantic segmentation using ADE20K as the downstream task. To ensure a fair comparison, both MAE and MLO-MAE were exclusively pretrained and fine-tuned on the ADE20K training dataset. The empirical results, presented in the table below, highlight MLO-MAE’s ability to effectively utilize diverse downstream tasks as feedback to guide the masking selection in Stage III, demonstrating its versatility and adaptability.
>
> |       Model       | mIoU (ADE20K) |
> |--------------------|---------------|
> | MAE               | 28.8          |
> | MLO-MAE           | 29.1          |
>
>
> ## 2. MLO-MAE's performance without pretraining on MAE
> We performed MLO-MAE sole pretraining (without using ImageNet pretrained initialization) on the PDDB and PAD-UFES datasets. The results, presented below, demonstrate that MLO-MAE consistently outperforms the baseline methods even in this non-continued pretraining setting.
>
> | Method                                 | PDDB Acc(%) | PAD-UFES Acc(%) |
> |----------------------------------------|-------------|--------------|
> | No continued pretraining               | 88.6        | 75.0            |
> | MAE continued pretraining              | 89.3        | 75.4            |
> |**MLO-MAE pretraining (no IN pretrain)**   | **90.2**       | **75.9**            |
> | MLO-MAE continued pretraining          | 92.7        | 77.6            |
>
>
> As expected, the performance of MLO-MAE without ImageNet initialization is slightly lower compared to MLO-MAE with continued pretraining. This difference is due to the latter benefiting from pretraining on significantly larger and more diverse data, providing a stronger foundation for fine-tuning.
>
> ## 3. Discussion on convergence rate
> We thank the reviewer for this insightful question regarding the faster convergence of MLO-MAE compared to baseline methods.  MLO-MAE’s epoch structure differs fundamentally from baseline methods. Each epoch in MLO-MAE involves solving three interconnected optimization stages (pretraining image encoder, training classification head, and updating masking network). This end-to-end design ensures a tighter alignment between the pretraining objective and downstream performance, effectively reducing the number of epochs required to achieve strong results. In contrast, methods like MAE employ random masking and separate pretraining from downstream evaluation, often necessitating significantly more epochs for convergence.
>
> As an alternative, we have reported the training time of the baseline and MLO-MAE for comparison. As shown in Table 9, the training time of MLO-MAE that leads to convergence is significantly smaller than that of MAE, indicating the faster convergence rate of MLO-MAE. We also observe a consistent trend with other informed masking methods, such as SemMAE and AutoMAE, which converge in fewer epochs compared to vanilla MAE (800 vs. 1600 epochs).

---

### Review · Reviewer_n4Bm · 2025-01-09

**Summary Of Contributions:**

In the context of training Masked Autoencoders, the paper introduces a novel method based on Multi level optimization to enhance the effectiveness of the training. Unlike previous approaches based on independent strategies or heuristics, the presented method the MLO-MAE method tries to focus the most informative regions - using the feedback from a downstream task to evaluate this informativeness. The training process is in 3 optimization stages:
1. Pretraining an encoder using a tentative version of the mask
2. Training a classifier head using the representation learned by the encoder
3. Update the mask from step 1 with the performance of the classifier on a validation set

After presenting the method in details, the paper showcases its performance in a variety of settings, and discusses its benefits in terms of better masking and representation learning.

**Audience:**

Yes

**Claims And Evidence:**

Yes

**Requested Changes:**

See weaknesses sections.

**Strengths And Weaknesses:**

Strengths:
- Multi-level optimization formulation of the problem is novel and experimental results look promising - in particular the use of the performance on a downstream task as a feedback is an elegant idea
- End-to-end pipeline is an interesting a very generalizable approach
- In depth discussion of experimental results allows to get a good understanding of the behavior of the method in different settings and its properties (especially sections 3.5, 3.6, 3.7)

Weaknesses:
- One of the main issues with the paper is the lack of clarity: end-to-end pipeline is not explained well and it takes some time for the reader to understand the method
- Likewise, the paper can be better organized, e.g. section 3 is very broad - from method overview to experimental results discussion and property analyses.
- Lack of theoretical analysis of the optimization problem - would have liked an analysis of the optimality of the formulation, e.g. bounds or at least a discussion on the fact that it can be seen as an approximation approach to jointly solving the 3 stages at once
- No discussion on the potential limitations of the method

---

> ### Author Response · Authors · 2025-01-16
>
> ## 1. Lack of clarity
> We appreciate the reviewer's feedback on the clarity of our method's explanation. The MLO-MAE framework consists of three stages executed in an integrated manner: pretraining the image encoder by reconstructing masked patches, training a classification head using representations from the pretrained encoder, and refining the masking network based on feedback from downstream validation performance. These stages are interconnected through a multi-level optimization process, where the validation performance directly influences the masking strategy. To ensure our end-to-end pipeline is more comprehensible, we have revised and reorganized Section 3.1 and will be updated in the revised version.
>
> ## 2. Paper organization
> We thank the reviewer for the suggestion. We have separated the experimental results into a new section and will be updated in the revised version.
>
> ## 3. Discussion on the convergence
> Previous works have shown the convergence analysis of bilevel optimization[1,4,5] and such analysis can be readily extended to multi-level optimization. We have adopted several strategies to enhance the practical effectiveness and near-optimality of our formulation:
> - Our optimization algorithm utilizes efficient hypergradient-based updates with a finite number of unrolling steps to approximate solutions at each level. While these approximations do not guarantee global optimality, they ensure convergence to stable solutions that are effective in practice. This is consistent with prior works in multi-level optimization[2,3].
> - Conceptually, our framework can be seen as an approximation to solving the three stages jointly by iteratively refining the parameters of the masking network $T’\approx T^*$. This iterative approach, combined with our finite unrolling strategy, ensures that the downstream validation performance closely guides the pretraining and masking processes, thereby indirectly approximating the optimal joint solution.
>
> While our optimization algorithm does not guarantee convergence to the global optimal masking network $T^*$, our experimental results (Tables 1-6) demonstrate that MLO-MAE significantly outperforms baseline methods across diverse datasets and tasks. These empirical validations support the effectiveness of our approach in achieving near-optimal solutions for the joint problem.
> We will add further discussion in Appendix A.5 on the convergence property in the revised version.
>
> [1] iDARTS: Differentiable Architecture Search with Stochastic Implicit Gradients, ICML 2021
>
> [2] Betty: An Automatic Differentiation Library for Multilevel Optimization, ICLR 2023
>
> [3] Transformer Architecture Search for Improving Out-of-Domain Generalization in Machine Translation, TMLR 2024
>
> [4] On the Iteration Complexity of Hypergradient Computation, ICML 2020
>
> [5] Towards gradient-based bilevel optimization with non-convex followers and beyond, NeurIPS 2021
>
>
> ## 4. Discussion on limitation
> We appreciate the reviewer’s feedback and acknowledge that our method has certain limitations. One key limitation is the higher computational overhead introduced by the calculation of the hypergradient in the multilevel optimization process. To address this, we employ gradient approximation, where the level of approximation is controlled by adjusting the number of unrolling steps. This approach balances computational cost with optimization accuracy, ensuring practical applicability.
>
> Another limitation is the increased GPU memory usage, which arises from the inclusion of the classification head and the need to load separate training datasets for the three stages into the GPU simultaneously. While we have not explicitly addressed this challenge in our current implementation, potential solutions include the use of gradient accumulation, which allows for smaller batch sizes without compromising model performance. We believe such strategies could alleviate memory constraints and enhance the method’s accessibility for resource-limited environments.

---

### Review · Reviewer_Q4kg · 2025-02-15

**Summary Of Contributions:**

This paper introduces Multi-level Optimized Mask Autoencoder (MLO-MAE), a novel framework for improving the representation ability of MAE model by optimizing the masking strategy using the feedback from downstream tasks, such using image labels from the classification task. The author designs a multi-level optimization (MLO) framework to jointly learn multiple mutually-independent modules via an end-to-end three stage training pipeline.

**Audience:**

Yes

**Claims And Evidence:**

Yes

**Requested Changes:**

See the weaknesses.

**Strengths And Weaknesses:**

Advantages:
This paper includes labels from down-stream tasks into the training stage of the MAE encoder. This is well motivated since the random masking strategy is non-optimal to fit the distribution of down-stream data. This paper proposes to learn a masking generator network to generate mask according to specific data. The authors also formulate the proposed learning framework as a multi-level optimization problem, where each stage corresponds to one level of optimization. The three levels are mutually dependent and solved jointly.  This nested optimization structure enables the model to learn a masking strategy that is directly influenced by downstream task performance, rather than relying on a fixed or heuristic masking approach.


Weakness:
(1)	Dependency on Downstream Tasks. On the one hand, the masking network learned by MLO-MAE is heavily dependent on the specific downstream task used during pretraining. What if all the available data are unlabeled during the pretraining stage? Will this lack-of-label situation make the MLO-MAE algorithm non-usable? On the other hand, although the encoder network is not directly affected by the gradient generated by the image labels, the training of the masking network T is supervised by the labels. This indirect supervision from the labels implicitly boost the representation learning of the MAE encoder, making the comparisons with other baseline methods somewhat unfair.
(2)	High Computational Cost. The multi-level optimization introduces great computational complexity due to the nested optimization problems. Although the authors propose methods to mitigate this, the overall computational cost remains high compared to other MAE methods, which might limits the scalability of MLO-MAE to very large datasets or backbone models.

---

> ### Author Response · Authors · 2025-02-16
>
> We appreciate the reviewer’s insightful questions. Below, we address the concerns systematically.
> ## 1. Handling Fully Unlabeled Pretraining Data
> The reviewer raises a valid scenario where all pretraining data is unlabeled. Here, we would like to consider two scenarios: 1) using MLO-MAE in the continued pretraining stage, and 2) direct transfer learning from the MLO-MAE pretrained model. While MLO-MAE’s current formulation leverages downstream task labels to guide masking during pretraining, the framework can still operate in label-scarce settings through the following mechanisms:
> - MLO-MAE in the continued pretraining stage: MLO-MAE can be initialized with a vanilla MAE pretrained on unlabeled data. Subsequent MLO-MAE continued pretraining on the labeled data can be used to further improve the performance on downstream tasks (shown in Section 3.6), demonstrating MLO-MAE’s flexibility under diverse scenarios. Note that in this scenario, the continued pretraining is happening in the finetuning stage, therefore we can assume to have access to the downstream labeled data.
> - Direct Transfer Learning from MLO-MAE: When domain-specific labels are unavailable, MLO-MAE can leverage public labeled datasets (e.g., ImageNet) to learn generalizable masking strategies. First, we initialize MLO-MAE with a standard MAE pretrained on unlabeled domain-specific data. We then pretrain it on ImageNet using its labels to guide task-aware masking, and finally transfer the model to downstream tasks without requiring their labels during pretraining. Our experiments in Section 3.5 show that MLO-MAE pretrained on one labeled dataset (e.g., ImageNet) transfers effectively to unseen tasks/datasets without requiring their labels during pretraining. This confirms that the learned masking strategy generalizes beyond the specific downstream task used for guidance.
>
> ## 2. Fairness of Comparisons
> The reviewer questions whether indirect label usage in MLO-MAE biases comparisons with baselines. Here we would like to clarify:
> - Equivalent Label Utilization: All baseline methods (MAE, SemMAE, etc.) require labeled data for fine-tuning or linear probing post-pretraining. In MLO-MAE, the same labeled data is used only during pretraining (Stage II and Stage III) to guide masking, not for direct encoder updates. Thus, no additional labels are introduced—comparisons remain fair (Section 3.4).
> - Empirical Validation: Transfer learning results (Tables 3–5) demonstrate that MLO-MAE’s gains persist even when pretrained on one task (e.g., ImageNet classification) and evaluated on others (e.g., ADE20K segmentation). This confirms that performance improvements stem from better representation learning.
>
> In summary, MLO-MAE does not assume access to "extra" labels but reuses the same labels required by all methods for evaluation. Our design ensures fairness while enabling adaptability to label-scarce settings through continued pretraining and transfer learning.
>
> ## 3. Computational Cost
> We appreciate the reviewer's concern regarding computational complexity. While multi-level optimization inherently increases per-iteration costs due to nested updates, our framework achieves significant time savings through two key aspects:
> - Faster Convergence: As shown in Table 9, MLO-MAE requires only 50 epochs to outperform MAE (1,600 epochs) and SemMAE/AutoMAE (800 epochs). Despite higher per-epoch costs, the total pretraining time for MLO-MAE on ImageNet-1K is 1,083 GPU hours, which is 49% lower than MAE (2,132 hours) and comparable to AutoMAE (1,344 hours). This demonstrates that the efficiency gains from reduced epochs outweigh the per-iteration overhead.
> - Optimized Unrolling Steps: As discussed in Section 3.8 (Table 7), we empirically found diminishing returns with increased unrolling steps. By limiting unrolling steps to 2, we balance gradient accuracy and computational cost, ensuring scalability without sacrificing performance.
>
> The results on ImageNet-1K—a large-scale benchmark—validate MLO-MAE’s scalability. We acknowledge that further optimizations (e.g., distributed training) could enhance efficiency, but our current implementation already demonstrates practical viability for large datasets.

---

### Decision · Action_Editor_fyKD · 2025-03-18

**Recommendation:** Accept as is

**Comment:**

This paper introduces Multi-level Optimized Mask Autoencoder (MLO-MAE), a novel framework for improving the representation ability of MAE model by optimizing the masking strategy using the feedback from downstream tasks. The method is well-motivated since the random masking strategy is obviously not an optimal solution to fit the downstream data. To achieve this, the authors proposed an end-to-end pipeline and the experimental results are promising.

All reviewers support the novelty of the work. During the discussion period, the authors well addressed the issues related to training convergence and computational cost. I am willing to follow the majority as all reviewers lean toward acceptance.

**Audience:**

The method can be applied to improve the performance of image encoders, which is an important area of research in computer vision.

**Claims And Evidence:**

This paper introduces Multi-level Optimized Mask Autoencoder (MLO-MAE), a new framework for improving the representation ability of MAE model by optimizing the masking strategy using the feedback from downstream tasks. A set of experiments supports the method and claims.